

# Tropospheric ozone in CCMI models and Gaussian emulation to understand biases in the SOCOLv3 chemistry-climate model

Laura E. Revell[1,2,3], Andrea Stenke[2], Fiona Tummon[2,4], Aryeh Feinberg[2], Eugene Rozanov[2,5], Thomas Peter[2], N. Luke Abraham[6,7], Hideharu Akiyoshi[8], Alexander T. Archibald[6,7], Neal Butchart[9], Makoto Deushi[10], Patrick Jöckel[11], Douglas Kinnison[12], Martine Michou[13], Olaf Morgenstern[14], Fiona M. O'Connor[9], Luke D. Oman[15], Giovanni Pitari[16], David A. Plummer[17], Robyn Schofield[18,19], Kane Stone[18,19,20], Simone Tilmes[12], Daniele Visioni[16], Yousuke Yamashita[8,21], and Guang Zeng[14]

[1]Bodeker Scientific, Christchurch, New Zealand
[2]Institute for Atmospheric and Climate Science, ETH Zurich, Zurich, Switzerland
[3]School of Physical and Chemical Sciences, University of Canterbury, Christchurch, New Zealand
[4]Now at: Biosciences, Fisheries, and Economics Faculty, University of Tromsø, Norway
[5]Physical-Meteorological Observatory/World Radiation Center, Davos, Switzerland
[6]Department of Chemistry, University of Cambridge, Cambridge, UK
[7]National Centre for Atmospheric Science (NCAS), UK
[8]National Institute of Environmental Studies (NIES), Tsukuba, Japan
[9]Met Office Hadley Centre (MOHC), Exeter, UK
[10]Meteorological Research Institute (MRI), Tsukuba, Japan
[11]Institut für Physik der Atmosphäre, Deutsches Zentrum für Luft- und Raumfahrt (DLR), Oberpfaffenhofen, Germany
[12]National Center for Atmospheric Research (NCAR), Boulder, Colorado, USA
[13]CNRM UMR 3589, Météo-France/CNRS, Toulouse, France
[14]National Institute of Water and Atmospheric Research (NIWA), Wellington, New Zealand
[15]National Aeronautics and Space Administration Goddard Space Flight Center (NASA GSFC), Greenbelt, Maryland, USA
[16]Department of Physical and Chemical Sciences, Universitá dell'Aquila, L'Aquila, Italy
[17]Environment and Climate Change Canada, Montréal, Canada
[18]School of Earth Sciences, University of Melbourne, Melbourne, Victoria, Australia
[19]ARC Centre of Excellence for Climate System Science, University of New South Wales, Sydney, Australia
[20]Now at: Department of Earth, Atmospheric and Planetary Sciences, Massachusetts Institute of Technology (MIT), Cambridge, Massachusetts, USA
[21]Now at: Japan Agency for Marine-Earth Science and Technology (JAMSTEC), Yokohama, Japan

**Correspondence:** Laura Revell (laura@bodekerscientific.com)

**Abstract.** Previous multi-model intercomparisons have shown that chemistry-climate models exhibit significant biases in tropospheric ozone compared with observations. We investigate annual-mean tropospheric column ozone in 15 models participating in the SPARC/IGAC (Stratosphere-troposphere Processes and their Role in Climate/International Global Atmospheric Chemistry) Chemistry-Climate Model Initiative (CCMI). These models exhibit a positive bias, on average, of up to 40–50% in the Northern Hemisphere compared with observations derived from the Ozone Monitoring Instrument and Microwave Limb Sounder (OMI/MLS), and a negative bias of up to ~30% in the Southern Hemisphere. SOCOLv3.0 (version 3 of the Solar-Climate Ozone Links CCM), which participated in CCMI, simulates global-mean tropospheric ozone columns of 40.2 DU – approximately 33% larger than the CCMI multi-model mean. Here we introduce an updated version of SOCOLv3.0, "SO-





COLv3.1", which includes an improved treatment of ozone sink processes, and results in a reduction in the tropospheric column ozone bias of up to 8 DU, mostly due to the inclusion of $N_2O_5$ hydrolysis on tropospheric aerosols. As a result of these developments, tropospheric column ozone amounts simulated by SOCOLv3.1 are comparable with several other CCMI models. We apply Gaussian process emulation and sensitivity analysis to understand the remaining ozone bias in SOCOLv3.1.

This shows that ozone precursors (nitrogen oxides ($NO_x$), carbon monoxide, methane and other volatile organic compounds) are responsible for more than 90% of the variance in tropospheric ozone. However, it may not be the emissions inventories themselves that result in the bias, but how the emissions are handled in SOCOLv3.1, and we discuss this in the wider context of the other CCMI models. Given that the emissions data set to be used for phase 6 of the Coupled Model Intercomparison Project includes approximately 20% more $NO_x$ than the data set used for CCMI, further work is urgently needed to address the

challenges of simulating sub-grid processes of importance to tropospheric ozone in the current generation of chemistry-climate models.

## 1   Introduction

Ozone is a key trace gas in the atmosphere. In the stratosphere, it absorbs UV-B ($280<\lambda<320$ nm) radiation and thus protects life at the surface. However in the troposphere, where approximately 10% of the total atmospheric ozone burden resides, ozone

is a greenhouse gas and air pollutant, with adverse affects on human health and crop yields (Myhre et al., 2013; Stevenson et al., 2013; Silva et al., 2013, 2017). Approximately 90% of tropospheric ozone results from a series of photochemical reactions which are initiated by the reaction of $NO_x$ (nitrogen oxides, $NO_x = NO+NO_2$) and either CO (carbon monoxide), $CH_4$ (methane) or a NMVOC (non-methane volatile organic compound) (Denman et al., 2007). These ozone precursors are emitted from, amongst other sources, fossil fuel burning, industrial processes and agriculture. Ozone can also be transported from the

stratosphere in stratosphere-troposphere exchange (STE) events. Greenslade et al. (2017) calculate the mean fraction of total tropospheric ozone attributable to STE as 1-3%, but show that during individual STE events, over 10% of tropospheric ozone may be directly transported from the stratosphere.

Due to its ~22 day lifetime in the troposphere, ozone is subject to intercontinental transport (Auvray and Bey, 2005), and this is modulated by decadal climate variability (Lin et al., 2014). Ozone is lost from the troposphere either by dry deposition

or photochemical destruction.

Most chemistry-climate models (CCMs), which are used to understand chemistry-climate interactions and project future atmospheric composition, overestimate tropospheric ozone in the Northern Hemisphere compared with observations (Young et al., 2013; Parrish et al., 2014). In particular, version 3.0 of the SOCOL (Solar-Climate Ozone Links) CCM (Section 2.1) contains notable positive tropospheric ozone biases. Revell et al. (2015) identified that ozone concentrations in SOCOLv3.0

are up to 50% too high in the Northern Hemisphere mid-troposphere (500 hPa) compared with observations from the Tropospheric Emission Spectrometer (TES). The reasons underlying SOCOLv3.0's tropospheric ozone bias were not completely clear to Revell et al. (2015), who noted that, while SOCOLv3.0 could accurately simulate the general geographic distribution





of tropospheric ozone, the actual magnitude was wrong and likely to be "a source issue (that is, emissions), a sink issue ($HNO_3$ washout), or a combination of the two."

Staehelin et al. (2017) showed that the mean tropospheric ozone burden in SOCOLv3.0 is 413 Tg, which is approximately 80 Tg larger than the multi-model mean burdens reported for the ACCENT (Atmospheric Composition Change: the European

Network of Excellence) and ACCMIP (Atmospheric Chemistry and Climate Model Intercomparison Project) activities, of 337 and 336 Tg, respectively. Furthermore, SOCOLv3.0 overestimates both the tropospheric ozone production and destruction rates compared to the multi-model means from ACCENT and ACCMIP (Staehelin et al., 2017). While SOCOLv3.0's production rates are overestimated by 34% compared to ACCENT and 41% compared to ACCMIP, the destruction rates are overestimated only by 20% (ACCENT) and 31% (ACCMIP).

SOCOLv3.0 participated in phase 1 of the Chemistry-Climate Model Initiative (CCMI) (Eyring et al., 2013; Morgenstern et al., 2017), which is a joint activity of SPARC (Stratosphere-troposphere processes and their role in Climate) and IGAC (International Global Atmospheric Chemistry), and is the successor activity to phase 2 of the Chemistry-Climate Model Validation activity, CCMVal-2 (SPARC CCMVal, 2010). Unlike CCMVal-2, which focussed on stratospheric processes and composition, CCMI includes many models with comprehensive representations of the troposphere, and aims to additionally address aspects

of tropospheric chemistry and circulation. Here, we examine tropospheric column ozone in SOCOLv3.0 and 14 other CCMI models in Section 3.1.

Recently a newer version of SOCOL has been developed, "SOCOLv3.1", which remediates obvious deficiencies in SOCOLv3.0's representation of tropospheric processes (Section 2.2). We compare tropospheric column ozone in SOCOLv3.0 and 3.1 with observations derived from OMI/MLS, the Ozone Monitoring Instrument/Microwave Limb Sounder, and use Gaussian

process emulation and sensitivity analysis to investigate the remaining ozone bias in SOCOLv3.1 (Section 3.2). Gaussian process emulation and variance-based sensitivity analysis allows a suite of model input parameters, and their relationship to the variable of interest, to be evaluated over a range of uncertainties of the inputs simultaneously at low computational cost. Here, we apply it to the SOCOLv3.1 tropospheric ozone budget to understand causes of the bias. In contrast to one-at-a-time testing, which investigates the model response to varying one input parameter while holding all others constant, Gaussian emulation

allows all parameters to be evaluated simultaneously and covers more of the parametric uncertainty space than one-at-a-time testing. Gaussian emulation is computationally efficient and allows the non-linear effects of the uncertainties on different input parameters to be accounted for. It also generates much more information than one-at-a-time testing – typically the same level of information as a Monte Carlo approach, but requiring a fraction of the model simulations (O'Hagan, 2006). Within the global atmospheric modelling community, Gaussian emulation has previously been applied to cloud and aerosol microphysics

modelling (Lee et al., 2011, 2012; Carslaw et al., 2013; Johnson et al., 2015).





## 2 Computational and statistical methods

### 2.1 The SOCOLv3.0 chemistry-climate model

The SOCOL model was developed in Switzerland at ETH Zurich and PMOD/WRC (the Physical Meteorological Observatory Davos/World Radiation Center). Version 3.0 of SOCOL (Stenke et al., 2013; Revell et al., 2015) consists of the middle atmo-
sphere version of the ECHAM5 (European Centre Hamburg Model) atmosphere-only general circulation model (Roeckner et al., 2003) coupled to the MEZON (Model for Ozone Trends) chemistry transport model (Egorova et al., 2003). The chemical solver takes into account 41 chemical species, 140 gas-phase reactions, 46 photolysis reactions and 16 heterogeneous reactions. The oxidation of isoprene, an important NMVOC for the tropospheric ozone budget, is accounted for with the Mainz Isoprene Mechanism (MIM-1), which comprises 16 organic degradation products of isoprene and a further 44 chemical reac-
tions (Pöschl et al., 2000). Global isoprene emissions are estimated to range from 440 to 660 Tg(C)/yr, which is comparable to the annual amount of $CH_4$ emissions (Guenther et al., 2006). About two thirds of the annual global emissions of volatile organic compounds (VOCs) are accounted for in SOCOLv3.0 by isoprene and methane. Apart from isoprene and formaldehyde, other NMVOCs are not included explicitly in the model but their contribution to CO is accounted for via the addition of a certain fraction of NMVOC emissions to CO. For anthropogenic, biomass burning and biogenic NMVOC emissions the
conversion factors to CO are 1.0, 0.31 and 0.83, respectively (Ehhalt et al., 2001).

Clear-sky photolysis rates are calculated online using a look-up-table (LUT) approach, which provides photolysis rates as a function of overhead ozone and oxygen columns (Rozanov et al., 1999). Variability of solar irradiance is included in the LUTs. Cloud impacts on photolysis are accounted for in the troposphere by the inclusion of a cloud modification factor following the parametrization described by Chang et al. (1987). From a recent intercomparison of photolysis rates simulated by different
CCMI models we learned that SOCOLv3.0 overestimates tropospheric $NO_2$ photolysis by roughly a factor of 2 compared to other models (Nicely et al., 2018). This overestimation is likely related to the treatment of backscattering from clouds in the calculations of the photolysis LUTs and the missing impact of aerosols. Both effects cannot be easily corrected by the implemented cloud modification factor, and so an online photolysis scheme is planned for future model versions.

Dry deposition velocities of $O_3$, CO, NO, $NO_2$, $HNO_3$ and $H_2O_2$ are based on Hauglustaine et al. (1994). This simplified
approach assumes constant dry deposition velocities over land and ocean, without accounting for seasonal or geographical variability. The tropospheric wash-out of $HNO_3$ and $H_2O_2$ is calculated by using a constant removal rate of $4 \times 10^{-6}$ s$^{-1}$, irrespective of precipitation occurrence. At every chemical time step, i.e., every two hours, 2.8% of tropospheric $HNO_3$ and $H_2O_2$ below 160 hPa are removed. Boundary conditions for the ozone precursor gases $NO_x$, CO and NMVOCs are implemented as surface emission fluxes. Methane is prescribed as a global average surface mixing ratio. For this study, both SOCOL
configurations were run with 39 vertical levels between Earth's surface and 0.01 hPa (∼80 km) and T42 horizontal resolution (grid cells approximately 2.8° × 2.8°).



## 2.2 Upgraded model version SOCOLv3.1

SOCOLv3.1 was developed to address SOCOLv3.0's representation of processes relevant to tropospheric ozone chemistry, with the aim of improving the model's large tropospheric ozone bias as shown by Revell et al. (2015).

First, we implemented heterogeneous hydrolysis of $N_2O_5$ on tropospheric aerosol, as this is an important removal process

for atmospheric $NO_x$ and was not included in SOCOLv3.0. As SOCOLv3.0 does not explicitly simulate tropospheric aerosols, the new scheme makes use of the ECHAM5 internal tropospheric aerosol climatology considering aerosol properties of 11 Global Aerosol Data Sets types (Köpke et al., 1997). The reaction probabilities for the different aerosol types are calculated following the parametrization by Evans and Jacob (2005).

Second, the simplified treatment of dry deposition was replaced by a more sophisticated scheme in SOCOLv3.1 based on

the surface resistances approach for the estimation of dry deposition velocities proposed by Wesely (1989). This takes into account actual meteorological conditions, different surface types and trace gas properties like solubility and reactivity. Further details of this scheme are given by Kerkweg et al. (2006).

Third, we adjusted how methane is prescribed in the model. In previous versions of SOCOL, methane was prescribed as a global surface average mixing ratio on the six lowermost model levels (covering approximately 2.5 km). This was changed to

only the surface level in SOCOLv3.1. While the amount of methane entering the atmosphere is the same in both configurations, prescribing it on one level instead of six means that methane-induced ozone production in the mid-to-upper troposphere is reduced. Because SOCOLv3 has a high OH bias compared to the ACCMIP models (Staehelin et al., 2017), ozone production from methane oxidation is amplified by the continuous re-supply of methane due to the mixing ratio boundary condition when methane is prescribed on six levels instead of one. An interhemispheric gradient and seasonal cycle in methane have also been

implemented in SOCOLv3.1, however these were not used in this study and instead methane was prescribed as a global average surface mixing ratio to test the general sensitivity of tropospheric ozone to surface methane concentrations.

Finally, because the LUTs used in SOCOLv3.0 cause tropospheric $NO_2$ photolysis to be overestimated due to the treatment of backscattering from clouds (Section 2.1), we recalculated LUTs for SOCOLv3.1. While the SOCOLv3.0 LUTs were calculated assuming 0.5 cloud coverage and a surface albedo of 0.3, the SOCOLv3.1 LUTs were based on clear-sky conditions and

also used a surface albedo of 0.3.

## 2.3 CCM simulations to compare with observations

We use the ensemble mean of three free-running SOCOLv3.0 simulations of the recent past to compare with observations (ETH-PMOD, 2015). These simulations were performed for CCMI, and conform to REF-C1 specifications (Eyring et al., 2013). The simulations cover the period 1960–2010, following a 10-year spin-up period. Greenhouse gas concentrations ($CH_4$,

$CO_2$ and $N_2O$) follow observations until 2005, then Representative Concentration Pathway (RCP) 8.5 (Riahi et al., 2011). Ozone precursor emissions (including $NO_x$, CO and NMVOCs) follow a historical emissions inventory until 2000 (Lamarque et al., 2010), then RCP 6.0 (Masui et al., 2011). Sea surface temperatures and sea ice concentrations were prescribed following HadISST observations (Rayner et al., 2003).





We also examine annual-mean tropospheric ozone in REF-C1 simulations performed by models participating in CCMI, described by Morgenstern et al. (2017) and references therein. Using the simulated ozone volume mixing ratio and WMO-defined tropopause height from each model, tropospheric ozone columns were calculated for the year 2005 by integrating ozone between the surface and WMO-defined tropopause. The WMO definition of the tropopause was selected to be consistent

with the OMI/MLS tropospheric ozone product (Ziemke et al., 2006). Between 2010–2014, the average tropospheric ozone burden derived from OMI/MLS was 300 Tg, which is very close to the multi-instrument mean of five satellite products over the same period, of 301 Tg (Gaudel et al., 2018).

Where multiple ensemble members ('realisations') of the REF-C1 simulation were submitted to the CCMI archive, the ensemble mean is shown. The exception is NIWA-UKCA, which submitted three realisations of the REF-C1 simulation,

however only the first realisation is shown as ozone precursor emissions were erroneously fixed at 1960 levels for the other two realisations (Morgenstern et al., 2017). The EMAC simulations used road traffic emissions which were updated every year rather than every month. Therefore when we examine year 2005 tropospheric column ozone in Section 3.1, the EMAC simulations used road traffic emissions for August 1954. Jöckel et al. (2016) show that this error results in tropospheric ozone columns that are ~2 DU lower than if the correct emissions were used. The UMUKCA-UCAM simulations used CCMVal-2

REF-B2 emissions for $NO_x$ aircraft emissions and $NO_x$, CO and HCHO surface emissions.

## 2.4    SOCOLv3.1 simulations for emulator training and testing

Variance-based sensitivity analysis allows the individual contribution of a single parameter to the overall uncertainty to be quantified. Because the large number of model simulations required would make one-at-a-time testing computationally too expensive, a Gaussian emulator can be used to supplement a complex model with a statistical model (Le Gratiet et al., 2017).

The output variable of interest (here tropospheric column ozone) is fitted with a mean function, and uncertainties are calculated with a covariance function, assuming that each unknown output point has a normal distribution. Non-linear contributions to the overall uncertainty in tropospheric column ozone can be identified by comparing the main effect variance (the reduction in the ozone variance when a particular model forcing is fixed, e.g. $NO_x$ emissions), with the total effect variance (the remaining variance in the tropospheric column ozone when everything except a particular model forcing is fixed). Various software

packages are available for Gaussian emulation. We used the Gaussian Emulation Machine for Sensitivity Analysis (GEM-SA), available at http://tonyohagan.co.uk/academic/GEM/index.html, to build an emulator for tropospheric column ozone.

Although many factors influence the tropospheric ozone budget, we restricted our analysis to 9 model forcings/parametrizations (see Table 1 for details of the scalings applied). These include:

1. $NO_x$ emissions (Denoted in figures as '$NO_x$').

2. Surface methane concentrations ('$CH_4$').

3. CO+NMVOC emissions ('CO').

4. the number of vertical levels $NO_x$ and CO+NMVOC emissions were prescribed on in the model ('ELEV').





5. the number of vertical levels $CH_4$ concentrations were prescribed on in the model ('CLEV').

6. the impact of clouds on photolysis rates, via the cloud modification factor ('CMF').

7. the rate of $HNO_3$ washout ('$HNO_3$').

8. the $N_2O_5$ uptake coefficient, which represents the probability of $N_2O_5$ hydrolysis occurring ('$N_2O_5$').

9. the specific reactivities for ozone dry deposition ('$O_3DD$'), which are used to estimate the dry deposition velocity.

Variables (1-3) were selected due to their importance as tropospheric ozone precursors. SOCOL contains only two NMVOCs, isoprene and formaldehyde, and other NMVOCs are represented as additional CO in the model (Section 2.1), hence for (3), CO and NMVOC emissions were varied simultaneously. SOCOLv3.0 and its predecessors prescribed methane on the lowermost six model levels. This was changed to only the surface level in SOCOLv3.1, and parameters (4) and (5) were included in our
analysis to investigate the sensitivity of tropospheric ozone to this implementation for all ozone precursors. By doing so, we aim to test the exchange of emissions between the boundary layer and free troposphere. The lowermost level in SOCOL covers approximately 100 m, and the 6 lowermost levels combined cover approximately 2.5 km. Parameter (6) was chosen because ozone production and destruction reactions are mostly photochemical, i.e. they occur in the presence of sunlight, and to test the sensitivity of the current CMF parametrization. Parameter (7) was selected because $HNO_3$ washout is the main sink for $NO_x$,
and therefore affects the ozone budget. Parameter (8) is similarly important as heterogeneous $N_2O_5$ hydrolysis leads to $HNO_3$ formation. Parameter (9) was chosen to test the sensitivity of tropospheric ozone to the newly-implemented dry deposition parametrization (Section 2.2).

90 SOCOLv3.1 "training" simulations (i.e. $10n$, where $n$ is the number of input parameters under investigation) were performed, and the resulting annual-mean tropospheric ozone column was used to construct the Gaussian process emulator in
several geographical regions (Europe, United States, Asia and the Southern Ocean). For each of the 90 training simulations, the 9 input variables were scaled simultaneously, with the scaling factors determined using a statistical method called a "maximin" Latin hypercube approach, which generates a near-random sample of parameter values from a multidimensional distribution while also maximising the uncertainty space (McKay et al., 1979). Table 1 summarises the minimum and maximum scalings applied to each of the 9 variables. These selected ranges are not necessarily feasible, but were selected to cover a range of
parametric uncertainties. This is discussed further in Section 3.2. Figure 1 shows the experimental design for the 90 training simulations, and is explained in further detail by Supplementary Video 1.

SOCOLv3.1 training simulations were performed for the year 2005 (following a common model spin-up period of 10 years, which was discarded from our analysis). The feedback between chemistry and radiation was switched off to keep internal variability as small as possible. Switching off the chemistry–radiation feedback means that all simulations have the same
meteorology (given that they started from the same initial conditions and ran with the same dynamical boundary conditions), despite having different chemistry. Therefore, we can be confident that the differences between the simulations, for example due to tropospheric ozone, are caused by differences in chemistry and not dynamics.





The emulator was constructed using tropospheric ozone columns calculated between the surface and the WMO-defined tropopause. We focus on four regions, namely Europe (37-60° N, 0-42° E), the United States (32-52° N, 67-124° W), Asia (6-49° N, 70-146° E) and the Southern Ocean (45–60° S, all longitudes), where different chemical regimes may dominate, e.g. Sillman et al. (1990).

After constructing the emulator, the next step is to validate it by comparing emulator-predicted ozone with SOCOL-simulated ozone. This was done by performing a further 27 (i.e. 3*n*) SOCOLv3.1 "testing" simulations. The set-up for these simulations was similar to the training simulations, with a new Latin hypercube generated to supply the scaling factors.

## 3   Results

### 3.1   Tropospheric ozone in the CCMI models

Figure 2 shows tropospheric ozone in the CCMI models, and illustrates the diversity amongst the models. Despite most of the models using ozone precursor emissions following the REF-C1 recommendations (see Section 2.3), they simulate vastly different representations of tropospheric ozone. A few of the models are closely related; for example the CESM1 models, WACCM and CAM4-chem, are essentially the same model in terms of tropospheric ozone. They differ only in the height of the model lid, which is 140 km for WACCM and 40 km for CAM4-Chem. ACCESS and NIWA-UKCA can also be considered the

same model for the REF-C1 experiment; although a coupled ocean was used for most of NIWA-UKCA's CCMI simulations, for the REF-C1 experiment they used the same prescribed sea surface conditions (temperature and ice coverage) as ACCESS. Differences between ACCESS and NIWA-UKCA in the REF-C1 simulation, therefore, are likely related to issues with the different compilers used which may induce small differences in stochastic physics and tropospheric age of air (Dietmüller et al., 2018).

The EMAC L47 and L90 models are also very similar; both have a model lid at 0.01 hPa (∼80 km), but they differ in the number of model levels between the surface and 0.01 hPa (47 and 90, respectively). They also use different time steps. Interestingly, EMAC-L90 simulates a better representation of tropospheric column ozone than EMAC-L47, despite the fact that EMAC-L90 has three fewer model levels between the surface and 300 hPa than EMAC-L47 and a longer time step. The difference in tropospheric column ozone between the two models likely results from the increased vertical resolution around

the tropopause in EMAC-L90, which has 11 levels between 300–100 hPa compared with 7 in EMAC-L47, meaning that EMAC-L90 better simulates stratosphere-troposphere exchange.

   Figure 3 shows the difference in tropospheric ozone between each of the CCMI models and OMI/MLS, and the root-mean-square error (RMSE) for the model-OMI/MLS difference. Alongside Fig. 2, Fig. 3 indicates clear outlying models in terms of tropospheric ozone. UMUKCA-UCAM simulates the smallest amount of tropospheric ozone (14.9 DU in the global mean,

Fig. 2o), however it only contains one NMVOC (formaldehyde) and does not 'lump' NMVOCs together in the way that many other CCMI models do. This means that additional NMVOC source gases are not considered by substituting with represented species, such as e.g. in SOCOLv3, whereby additional NMVOCs are included in the form of CO. Of the CCMI models, SOCOLv3.0 simulates the largest global-mean tropospheric ozone column, of 40.2 DU (Fig. 2a). SOCOLv3.0's tropospheric



ozone bias is investigated further in Section 3.2. In ULAQ-CCM, the zonal bands of large ozone abundances at northern and southern midlatitudes are related to the model's coarse horizontal resolution ($5.6° \times 5.6°$), which affects surface fluxes and tropospheric transport (Orbe et al., 2018).

Figure 4 shows multi-model means (MMM) and standard deviations. The MMM in Fig. 4a was calculated for all models, while the MMM in Fig. 4d was calculated only for models with a RMSE less than 10, as indicated in Fig. 3 – i.e., all models except SOCOLv3.0, ACCESS CCM, EMAC-L47, ULAQ-CCM and UMUKCA-UCAM. The CCMI models simulate a global-mean tropospheric ozone abundance of 31.1 DU (Fig. 4a), and 30.2 DU (Fig. 4d), depending on the MMM definition applied. Both global-mean MMMs are close to the OMI/MLS global mean of 28.6 DU (Fig. 5b). However, the MMMs differ markedly from OMI/MLS in terms of the global tropospheric ozone distribution.

Compared to OMI/MLS, the models overestimate tropospheric column ozone almost everywhere between 60° N–60° S (the region where OMI/MLS data are available), regardless of the MMM definition. The exception is at southern midlatitudes, where the models underestimate tropospheric ozone compared to OMI/MLS. When the MMM is calculated for all models, the positive bias is up to 50%, and the negative bias reaches up to -33% (Fig. 4c). When models with an RMSE>10 are discarded from the MMM, the negative bias is largely unchanged at -32%, but the positive bias is reduced, and reaches up to 40% (Fig. 4f). These results broadly agree with models evaluated as part of ACCMIP (Young et al., 2013), and phase 5 of the Coupled Model Intercomparison Project (CMIP5) (Eyring et al., 2013). The ACCMIP models used the same ozone precursor emissions as for CCMI and simulated, on average, up to 30% more tropospheric column ozone compared with OMI/MLS at northern midlatitudes (Young et al., 2013). For the CHEM models participating in CMIP5 (those models with interactive chemistry, i.e. ozone was calculated online and not prescribed from a climatology), the climatological-mean annual-mean MMM averaged over 2000-2005 was 30.5 DU (Eyring et al., 2013), which is similar to the MMMs calculated here. The CMIP5 and ACCMIP MMMs also show a stronger interhemispheric gradient than OMI/MLS observations do, consistent with our findings.

The standard deviation on the MMM is up to 11.3 DU when calculated for all models (Fig. 4b), and reduces to a maximum of 9.5 DU when calculated for only the "RMSE<10" models (Fig. 4e). The variability between models is largest at northern midlatitudes, and in the continental outflow region off the west coast of Africa.

## 3.2 Gaussian emulation and sensitivity analysis in SOCOLv3.1

Figure 5 compares annual-mean tropospheric column ozone as simulated by SOCOLv3.0 and 3.1 with observations derived from OMI/MLS. Although SOCOLv3.0 captures the spatial distribution of tropospheric ozone fairly well in a qualitative sense, i.e. elevated ozone in the Northern Hemisphere and a minimum over the tropical Western Pacific (Fig. 5a), it overestimates tropospheric column ozone between 60° N–40° S by up to 30 DU – approximately a factor of 2 (Fig. 5c). The improved treatment of ozone sink processes in SOCOLv3.1 means that tropospheric ozone columns are reduced globally by up to 8 DU compared with SOCOLv3.0 (Figs. 5d-e). Individual sensitivity tests (not shown) indicate that this is due mostly to the inclusion of heterogeneous $N_2O_5$ hydrolysis on tropospheric aerosol.

Both SOCOLv3.0 and 3.1 show a small negative bias in tropospheric ozone over the Southern Ocean. This was also visible in the SOCOLv3.0 and TES comparison presented by Revell et al. (2015). Recent work has indicated that the Wesely (1989)





dry deposition scheme overestimates the observed ozone deposition velocity by a factor of 2-4 in the Southern Ocean, where SSTs are low and chemical reactions are slow (Luhar et al., 2017). Further upgrades to the model's deposition scheme may therefore improve comparisons of simulated and observed tropospheric ozone in cold oceanic regions.

The global-mean tropospheric ozone column in SOCOLv3.1 is 36.4 DU (Fig. 5d), which is still at the upper end of the range of the CCMI models (Fig. 2), but comparable to other models such as ACCESS (36.3 DU), EMAC-L47 (37.3 DU) and MRI-ESMr1 (35.7 DU). Despite the improvements to SOCOLv3.1, a large bias in tropospheric ozone of approximately 20 DU compared with OMI/MLS remains (Fig. 5f). To understand the drivers of this remaining bias, we constructed a Gaussian emulator from the 90 SOCOLv3.1 "training" simulations (Section 2.4). Tropospheric ozone predicted by the emulator is compared with SOCOLv3.1 test simulations in Figure 6. In all four geographical regions shown, the correlation between emulated and simulated tropospheric ozone is high ($R^2 \geq 0.85$), indicating that the emulator performs well in these regions. The point with the largest simulated tropospheric ozone column corresponds to a simulation in which two ozone loss processes, $HNO_3$ washout and ozone dry deposition, were set to zero and large scalings (4.00 and 3.54) were applied to the ozone precursors $NO_x$ and $CH_4$, respectively, following the Latin hypercube design (Fig. 1). The emulator underestimates tropospheric ozone for this point in all four regions, indicating that it may not be well constrained at the extreme ends of the parameter uncertainty space – noting however, that within the uncertainty range this point agrees with the 1:1 line in all regions except the Southern Ocean.

Figure 7 displays the sensitivity of global-mean tropospheric ozone to each parameter assuming all other parameters are held constant. Greater uncertainty is indicated where the lines diverge (appearing as a thicker line). Tropospheric ozone exhibits a strong sensitivity to its precursor gases (Fig. 7a-c), and while the correlation between $CH_4$ and CO+NMVOCs is approximately linear, for $NO_x$ there appears to be a saturation effect for scaling factors greater than one. In our calculations a uniform sampling distribution was applied when generating the Latin hypercube, which means that in 25% of our training simulations the $NO_x$ (and $CH_4$, CO and NMVOC) scaling factors are less than one, while in the other 75% of simulations they are larger than one.

To test whether the emulator may be biased due to the sampling distribution used, we calculated tropospheric column ozone as a function of $NO_x$ and CO+NMVOCs using the gradients in Fig. 7a and c. Assuming a uniform sampling distribution between 0 and 4, as per the Latin hypercube design used here, the sensitivity indices for $NO_x$ and CO+NMVOCs are 0.68 and 0.32, respectively. If we assume a piecewise uniform distribution, so that 50% of the points are between 0 and 1 and 50% are between 1 and 4, the sensitivity indices are 0.72 for $NO_x$ and 0.28 for CO+NMVOCs. That is, the differences are negligible, implying that the type of sampling distribution used doesn't bias the result. However, given the $NO_x$ saturation effect above one (Fig. 7a), if we assume a uniform distribution between 0 and 2 instead of 0 and 4, the $NO_x$ sensitivity index increases to 0.86, while the CO index decreases to 0.14. This shows the importance of selecting an appropriate range for the parameter uncertainty space. However, the conclusions of our emulator analysis – that ozone precursors are the dominant driver of tropospheric ozone variability – remain unchanged.

Figure 8 shows the percentage of variance that each parameter contributes to in each geographic region, either jointly or alone. In all four regions, ozone precursors – $CH_4$, $NO_x$, CO and NMVOCs – account for 91-94% of the variance in tropospheric column ozone. In other words, changing these ozone source input parameters has a far larger impact on tropospheric



ozone abundances than changing ozone sink parameters does, and this applies to both polluted regions (Europe, the United States and Asia) and relatively pristine environments (the Southern Ocean). $NO_x$ emissions are generally the dominant driver of variability (in the European region they are approximately equal to the contribution from $CH_4$, Fig. 8a). Joint interactions between $NO_x$, $CH_4$ and CO+NMVOCs play a relatively minor role compared with the individual influences of these species.

Although updating SOCOLv3.1 with regards to $N_2O_5$ hydrolysis, $HNO_3$ washout, LUTs and ozone dry deposition results in a reduction in tropospheric ozone of 8 DU (Fig. 5e), as drivers of tropospheric ozone variability in SOCOLv3.1 they are insignificant compared with ozone precursors. However, we cannot discount the possibility that it is not the ozone precursor emissions themselves that are responsible for SOCOLv3's tropospheric ozone bias, but rather the way in which the emissions are handled by the model; this is discussed further in the following section.

**4   Discussion and conclusions**

Despite using the ozone precursor emissions recommended for CCMI, SOCOLv3.0 simulates the largest global-mean tropospheric ozone abundance of all the CCMI models (Fig. 2), and exhibits a bias of ∼30 DU compared with OMI/MLS observations (Fig. 5a). The CCMI MMM is biased high in the Northern Hemisphere and low in the Southern Hemisphere compared with OMI/MLS, consistent with previous studies relying on the same emissions inventories (Fig. 4c and f). We have developed

a new model version, SOCOLv3.1, which includes an upgraded treatment of tropospheric ozone sink processes. This results in a reduction in tropospheric ozone of up to 8 DU (Fig. 5e), which is mostly due to the inclusion of $N_2O_5$ hydrolysis on tropospheric aerosol. SOCOLv3.1 still exhibits a positive bias in tropospheric column relative to OMI/MLS (particularly in the Northern Hemisphere), but simulates tropospheric column ozone amounts that are much more comparable with the other CCMI models.

We have quantified the contribution to tropospheric ozone variance in SOCOLv3.1 from 9 model forcings/parametrizations using Gaussian process emulation and sensitivity analysis. By switching off the coupling between chemistry and radiation in the emulator experiments, we aimed to limit dynamical and meteorological variability. We did not consider stratosphere-troposphere exchange in our emulator experiments. Staehelin et al. (2017) showed that SOCOLv3.0's ozone burden due to stratospheric influx, when calculated from ozone origin tracers as described by Garny et al. (2011) and Revell et al. (2015), is

close to the multi-model mean values from the ACCMIP and ACCENT ensembles. Therefore, STE is unlikely to be a major driver of SOCOLv3's tropospheric ozone bias.

Our Gaussian process emulation experiments and sensitivity analysis illustrate that the ozone precursors $NO_x$, $CH_4$, CO and NMVOCs are responsible for more than 90% of the variance in tropospheric column ozone in the improved model version, SOCOLv3.1. While $CH_4$ is prescribed as a surface mixing ratio, the other ozone precursors are specified from emissions

inventories. Collating emissions inventories is challenging as they are typically compiled using a bottom-up approach. Anthropogenic emissions must rely on accurate reporting, while for biogenic emissions there are no reporting requirements. Furthermore, emissions are generally prescribed in global models as monthly means, and thus do not reflect diurnal or weekly variability (Young et al., 2018). Hassler et al. (2016) identified that current global emissions inventories do not capture trends



in the $NO_x$/CO ratio, and previous multi-model studies have also identified potential deficiencies with the inventories (Young et al., 2013; Parrish et al., 2014). Jena et al. (2015) and Zhong et al. (2016) showed that different $NO_x$ emissions inventories can significantly alter simulated tropospheric ozone.

However, it may not be the emissions used for CCMI themselves that are incorrect, but rather problems in how they are handled in global models. Given the coarse grid sizes necessary to run a global model and still retain computational efficiency, resolution – horizontal, vertical and temporal – is likely important for simulating tropospheric ozone, especially in polluted regions where very large emissions in an urban environment may be spread over a model grid cell spanning thousands of square kilometers. In global models, polluted air coming from a point source is considered to be well-mixed throughout a large grid cell, which would generally lead to more efficient ozone production (Young et al., 2018). Horizontal and vertical resolution are difficult to test in an emulator sensitivity study as presented here, however by examining the CCMI models collectively (Morgenstern et al., 2017), we can derive some insights. For example, we note that GEOSCCM, HadGEM3-ES and the CESM1 models (CAM4Chem and WACCM), which simulate the smallest RMSEs relative to OMI/MLS (Fig. 3d,e,j,k), have fairly high horizontal resolution relative to other CCMs, of $2° × 2°$, $1.875° × 1.25°$ and $1.9° × 2.5°$ degrees, respectively. Of the models analysed in this study, HadGEM3-ES also has the largest number of levels in the troposphere (48). Similarly, tropospheric ozone in the EMAC model with 90 levels (EMAC-L90) compares better with observations than the 47 level version (EMAC-L47) (Fig. 3h,i), which may be due to a more realistic simulation of the ozone gradient across the tropopause (Section 3.1).

SOCOLv3.0 uses T42 horizontal resolution (approx. $2.8° × 2.8°$), which is also used by CCSRNIES MIROC 3.2 and EMAC. With 16 vertical levels, SOCOLv3.0 has the smallest number of vertical levels in the troposphere out of all the models analysed here, except CCSRNIES MIROC3.2, which has 15. CCSRNIES-MIROC3.2, CNRM-CM5-3 and CMAM do not include any NMVOCs, while SOCOLv3.0 includes only 2 NMVOCs – isoprene and formaldehyde. Models with complex NMVOC schemes tend to simulate tropospheric ozone favourably compared to OMI/MLS, such as the CESM1 models, with 19 NMVOCs, and GEOSCCM, with 13 explicit NMVOCs.

Another respect in which SOCOLv3.0 is an outlier amongst the CCMI models is its chemical time step of two hours. The other models analysed in this study have chemical time steps ranging from 6 minutes (CCSRNIES-MIROC3.2) to one hour (the models based on the UK Met Office Unified Model, i.e. HadGEM3-ES, NIWA-UKCA, ACCESS and UMUKCA-UCAM). In a sensitivity test, SOCOLv3.0's chemical time step was reduced to 15 minutes, which reduced the ozone burden in polluted urban areas by approximately 5 DU (not shown). To test how SOCOL responds to prescribing a surface mixing ratio of $NO_x$ rather than an emissions flux, we performed a further sensitivity simulation where surface $NO_2$ mixing ratios from the CESM1 WACCM REF-C1 simulation were prescribed instead of $NO_x$ emissions. This also resulted in a reduction of tropospheric ozone of up to 5 DU. In reality there is likely no single solution for reducing SOCOLv3.0's excessive tropospheric ozone bias, however assuming that the prescribed emissions are correct, then increasing the model's spatial and temporal resolution within the bounds of computational efficiency will likely reduce the bias.

We have shown the importance of ozone precursor emissions for simulating the tropospheric ozone budget with SOCOLv3.1. This is in line with the findings of Revell et al. (2015), who analysed three SOCOLv3.0 simulations for the period 1960-2100:



REF-C2 (based on RCP 6.0), SEN-C2-fEmis (NO$_x$, CO and NMVOC emissions fixed at constant 1960 levels) and SEN-C2-fEmis-fCH$_4$ (Similar to SEN-C2-fEmis but with surface methane concentrations also fixed at constant 1960 levels). They showed that future global ozone abundances are governed largely by changes in methane and NO$_x$, with methane causing an increase in tropospheric ozone that is approximately one-third of that caused by NO$_x$. Future work should investigate how

tropospheric ozone evolves in future under the various CCMI sensitivity scenarios in all CCMI models.

Finally, phase 6 of the Coupled Model Intercomparison Project (CMIP6) will use the emissions data set described by Hoesly et al. (2018). In this data set, year 2000 NO$_x$ emissions are ∼20% larger than the emissions used for CCMI (Lamarque et al., 2010). Therefore, simulated ozone biases by the current generation of CCMs will likely be amplified in CMIP6.

Given the results of our multi-model intercomparison as well as previous multi-model studies, our results highlight the need

for careful validation of emissions inventories used by global models. However, the way in which emissions are handled by the models also appears to result in biased ozone abundances, and further work is needed to address the challenges of simulating sub-grid processes of importance to tropospheric ozone, in SOCOLv3 as well as in other CCMI models.

*Data availability.* The data used here (except the CESM1 data) are held at the Centre for Environmental Data Analysis (CEDA, http://data.ceda.ac.uk/badc/wcrp-ccmi/data/CCMI-1/). CESM1 WACCM and CESM1 CAM4-chem data were downloaded from http://www.

earthsystemgrid.org. For instructions for access to both archives see http://blogs.reading.ac.uk/ccmi/badc-data-access. GEOSCCM data were provided directly by L. Oman to replace the GEOSCCM data currently held in the CEDA archive. SOCOLv3.1 data are available by contacting L. Revell.

*Competing interests.* The authors declare no competing interests.

*Acknowledgements.* We acknowledge the modeling groups for making their simulations available for this analysis, the joint WCRP SPARC/IGAC

Chemistry-Climate Model Initiative (CCMI) for organizing and coordinating the model data analysis activity, and the British Atmospheric Data Centre (BADC) for collecting and archiving the CCMI model output. The EMAC simulations were performed at the German Climate Computing Centre (DKRZ) through support from the Bundesministerium für Bildung und Forschung (BMBF). DKRZ and its scientific steering committee are gratefully acknowledged for providing the HPC and data archiving resources for this consortial project ESCiMo (Earth System Chemistry integrated Modelling). We acknowledge the UK Met Office for use of the MetUM. This research was partially supported

by the NZ Government's Strategic Science Investment Fund (SSIF) through the NIWA programme CACV. OM acknowledges funding by the New Zealand Royal Society Marsden Fund (grant 12-NIW-006). The authors wish to acknowledge the contribution of NeSI high-performance computing facilities to the results of this research. New Zealand's national facilities are provided by the New Zealand eScience Infrastructure (NeSI) and funded jointly by NeSI's collaborator institutions and through the Ministry of Business, Innovation and Employment's Research Infrastructure programme (https://www.nesi.org.nz). FT was supported by SNSF grant number 20F121_138017. ACCESS-CCM runs were

supported by Australian Research Council's Centre of Excellence for Climate System Science (CE110001028), the Australian Government's



National Computational Merit Allocation Scheme (q90) and Australian Antarctic science grant program (FoRCES 4012). The HadGEM3-ES simulations from the Met Office were supported by the Joint UK BEIS/Defra Met Office Hadley Centre Climate Programme (GA01101) and the European Commission's 7th Framework Programme StratoClim project (grant agreement 603557). CCSRNIES research was supported by the Environment Research and Technology Development Fund (2-1303 and 2-1709) of the Ministry of the Environment, Japan, and

5   computations were performed on NEC-SX9/A(ECO) computers at the CGER, NIES. UMUKCA-UCAM model integrations were performed using the ARCHER UK National Supercomputing Service and MONSooN system, a collaborative facility supplied under the Joint Weather and Climate Research Programme, which is a strategic partnership between the UK Met Office and the Natural Environment Research Council.



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



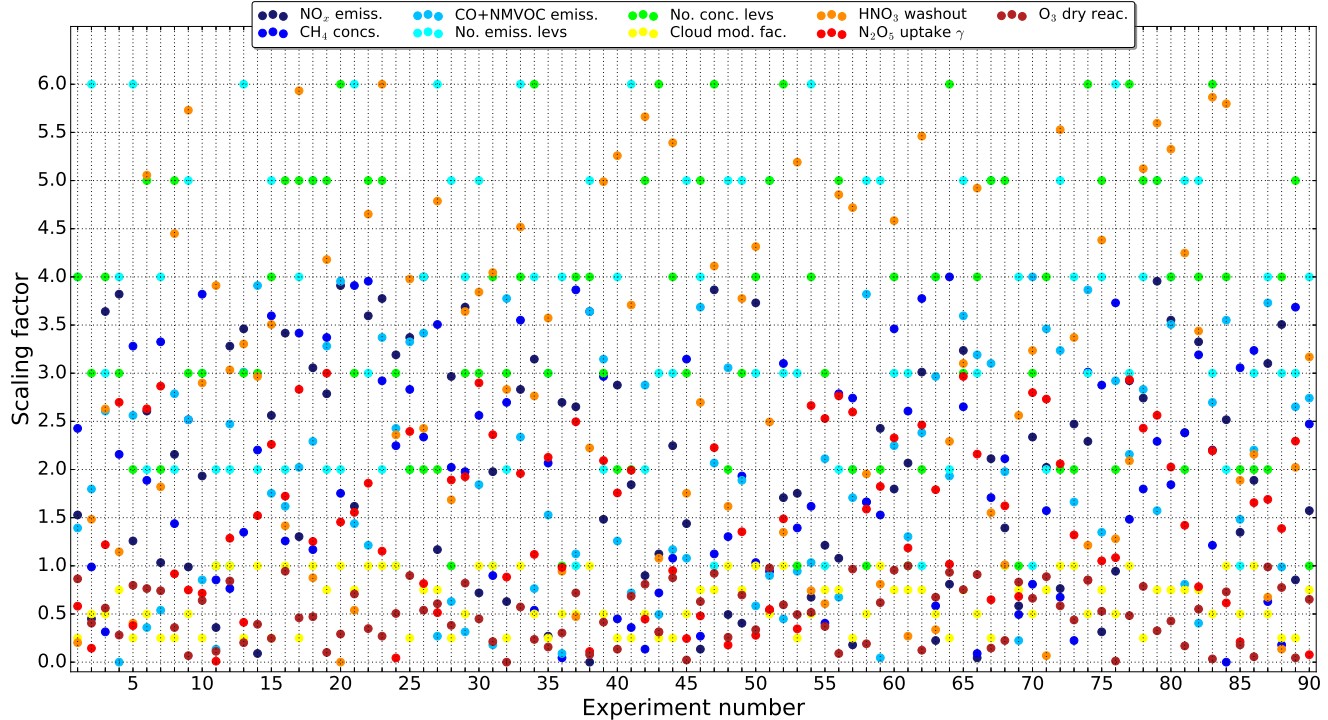

**Figure 1.** Experimental design for the 90 SOCOLv3.1 simulations performed to train the emulator. Each column of dots indicates the scaling applied to each of the 9 variables – see Table 1 for more details. For clarity the $N_2O_5$ hydrolysis scaling factors have been multiplied by 10 here, and the $HNO_3$ washout scaling factors have been multiplied by 12. See also Supplementary Video 1, which explains this figure in greater detail.





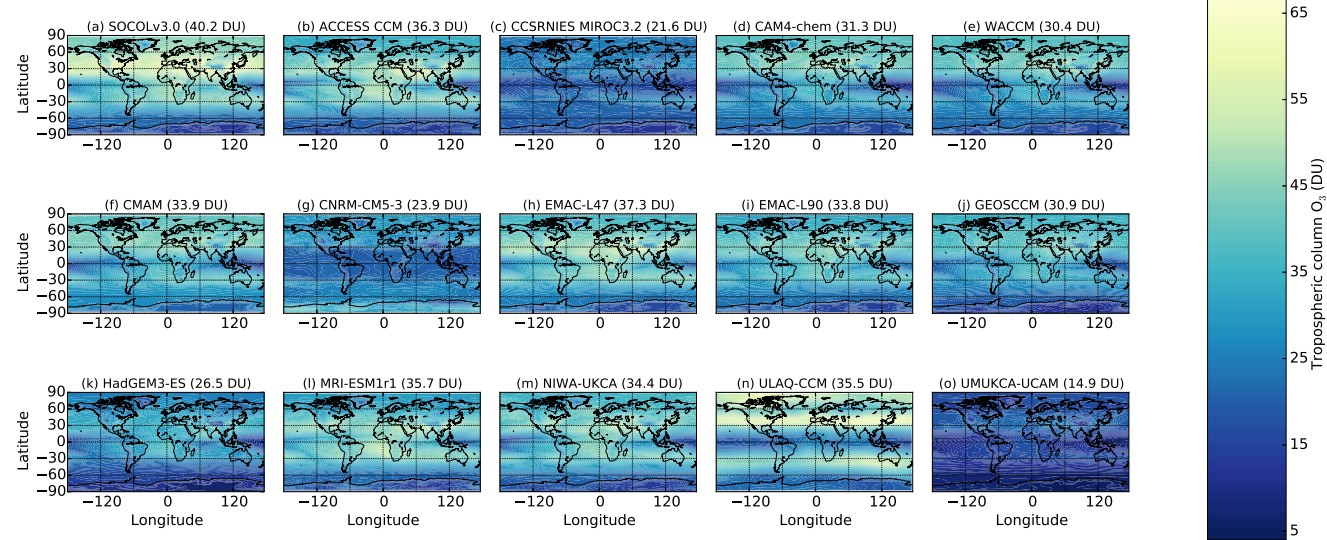

**Figure 2.** Annual-mean year 2005 tropospheric ozone columns in REF-C1 simulations from CCMI models (calculated relative to the WMO-defined tropopause pressure for each model). The global-mean tropospheric column ozone amount for each model is indicated in the title.





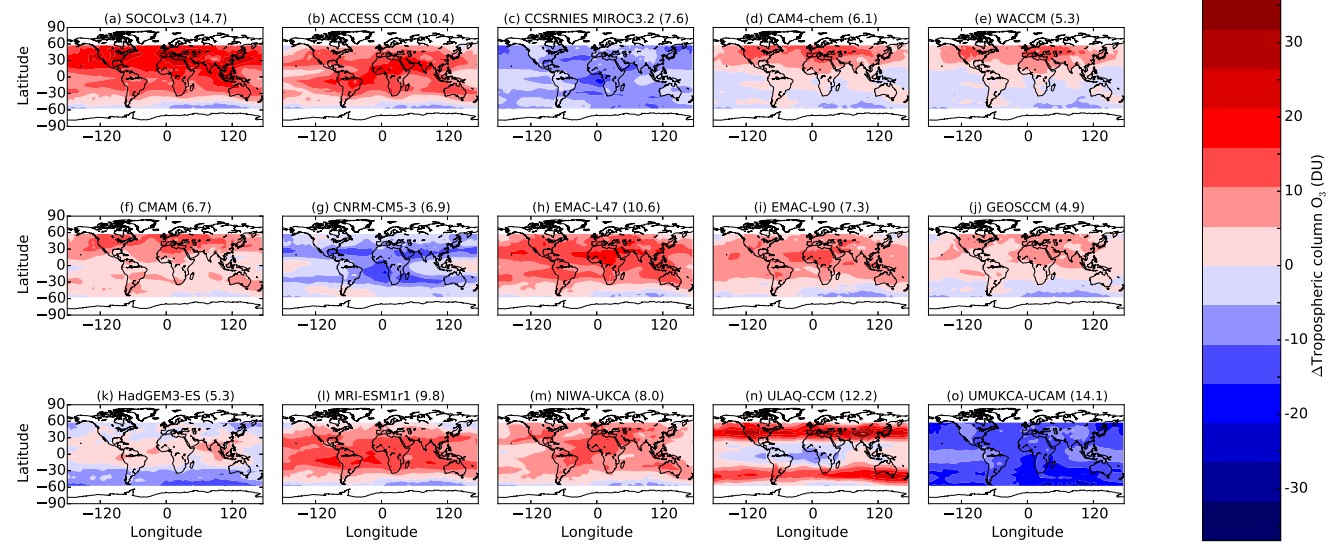

**Figure 3.** Difference between annual-mean year 2005 tropospheric column ozone in CCMI models compared with OMI/MLS, i.e. model minus OMI/MLS. The root-mean-square error for each model compared with OMI/MLS is indicated in the title.





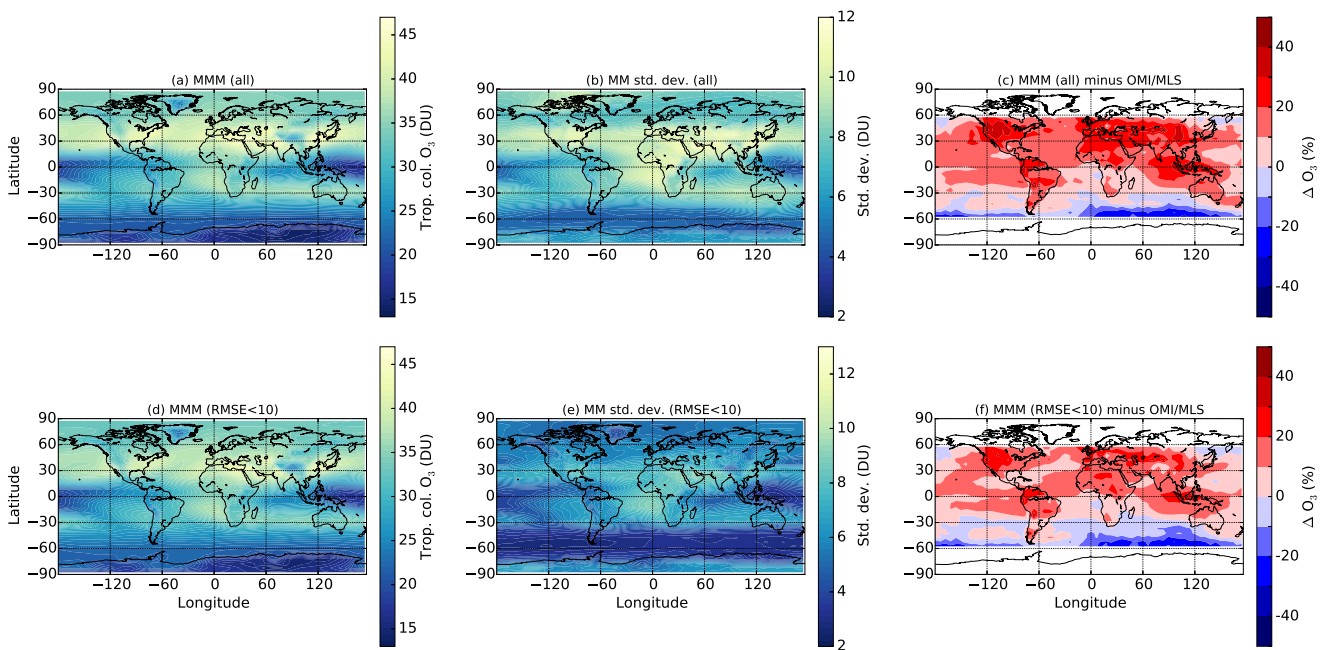

**Figure 4.** Annual-mean year 2005 tropospheric column ozone. (a) The multi-model mean (MMM) of all CCMI models; (b) multi-model standard deviation for the models shown in (a); (c) percent difference between the MMM in (a) and OMI/MLS (MMM minus OMI/MLS); (d) MMM for a subset of CCMI models – those with a root-mean-square error (RMSE) less than 10 when compared with OMI (see Fig. 3); (e) multi-model standard deviation for the models shown in (d); (f) percent difference between the MMM in (d) and OMI/MLS (MMM minus OMI/MLS).



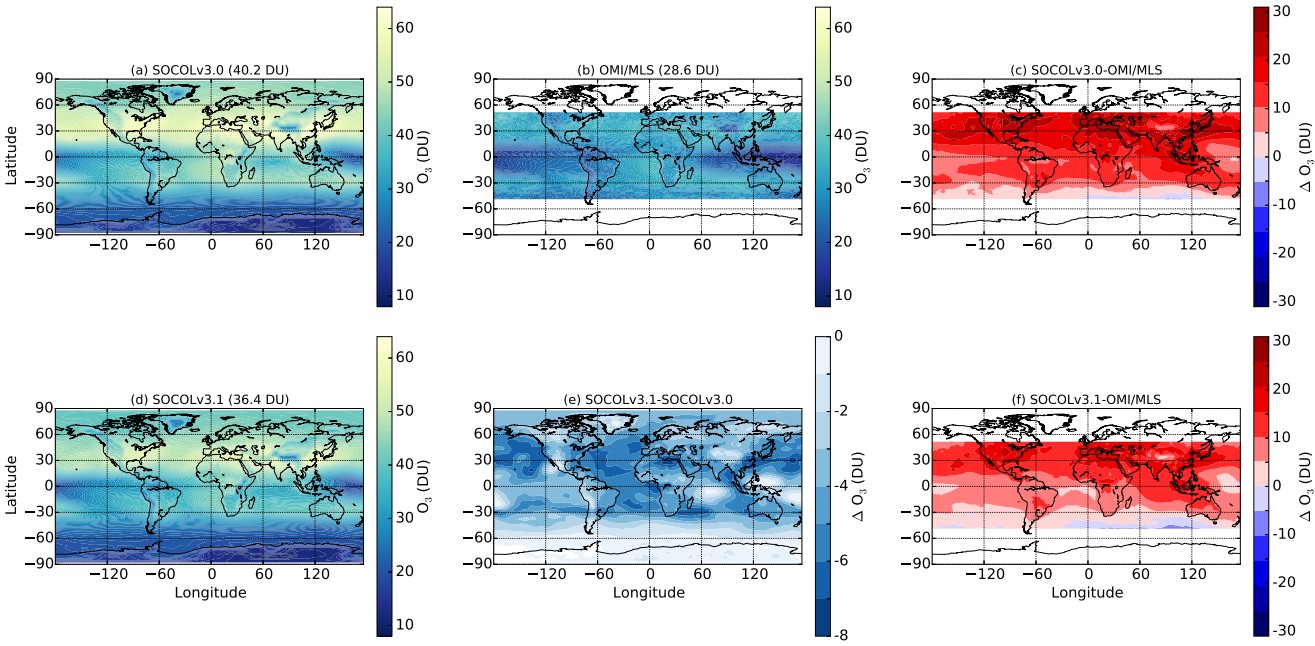

**Figure 5.** Annual-mean year 2005 tropospheric column for: (a) SOCOLv3.0; (b) OMI/MLS observations; (c) The difference between SOCOLv3.0 and OMI/MLS; (d) SOCOLv3.1; (e) The difference between SOCOLv3.1 and SOCOLv3.0; (f) The difference between SOCOLv3.1 and OMI/MLS. The global-mean tropospheric column ozone amount is indicated in the title for (a), (b) and (d).





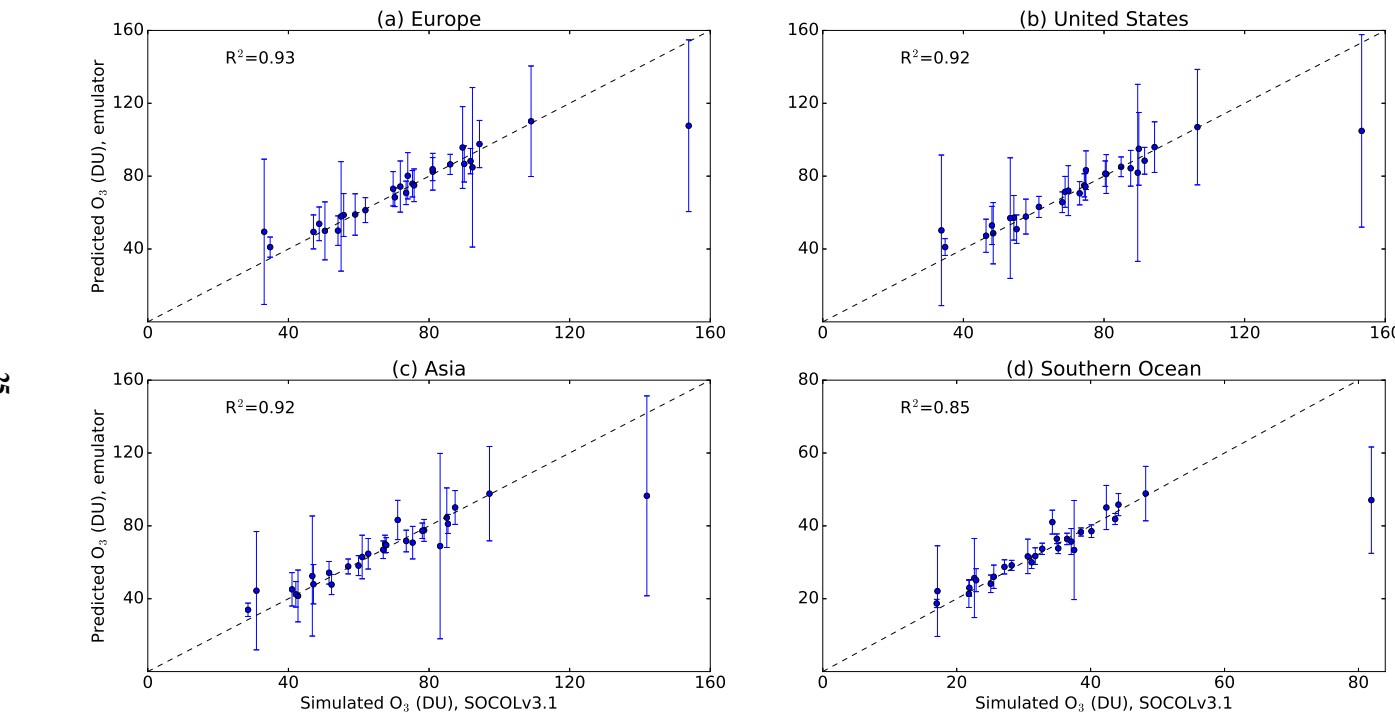

**Figure 6.** Tropospheric column ozone as predicted by the emulator, vs. the amount simulated in SOCOLv3.1 test simulations. The 1:1 line is also shown.



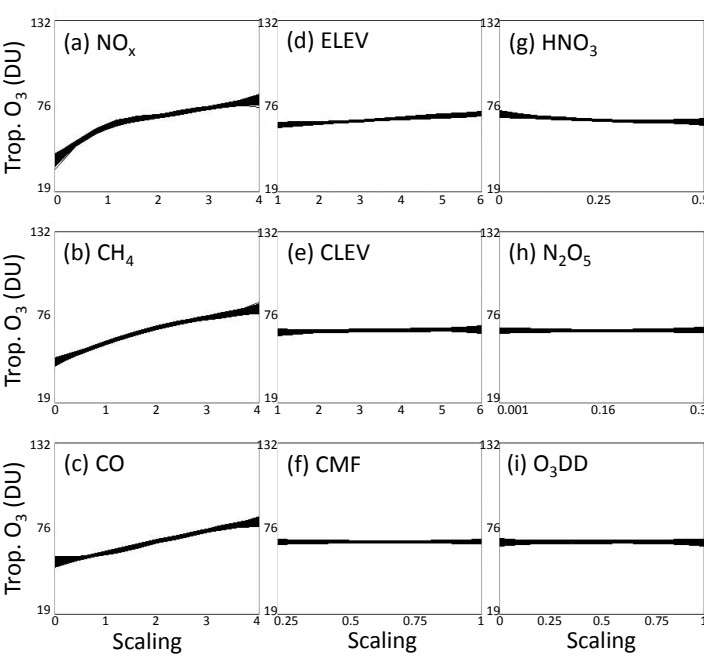

**Figure 7.** Sensitivity of annual-mean tropospheric column ozone in 2005 to each of the 9 variables listed in Table 1, assuming the other variables are constant. The horizontal axis shows the range of scaling factors applied to each variable. This plot is for the Asian region only, however it is representative of the other regions.



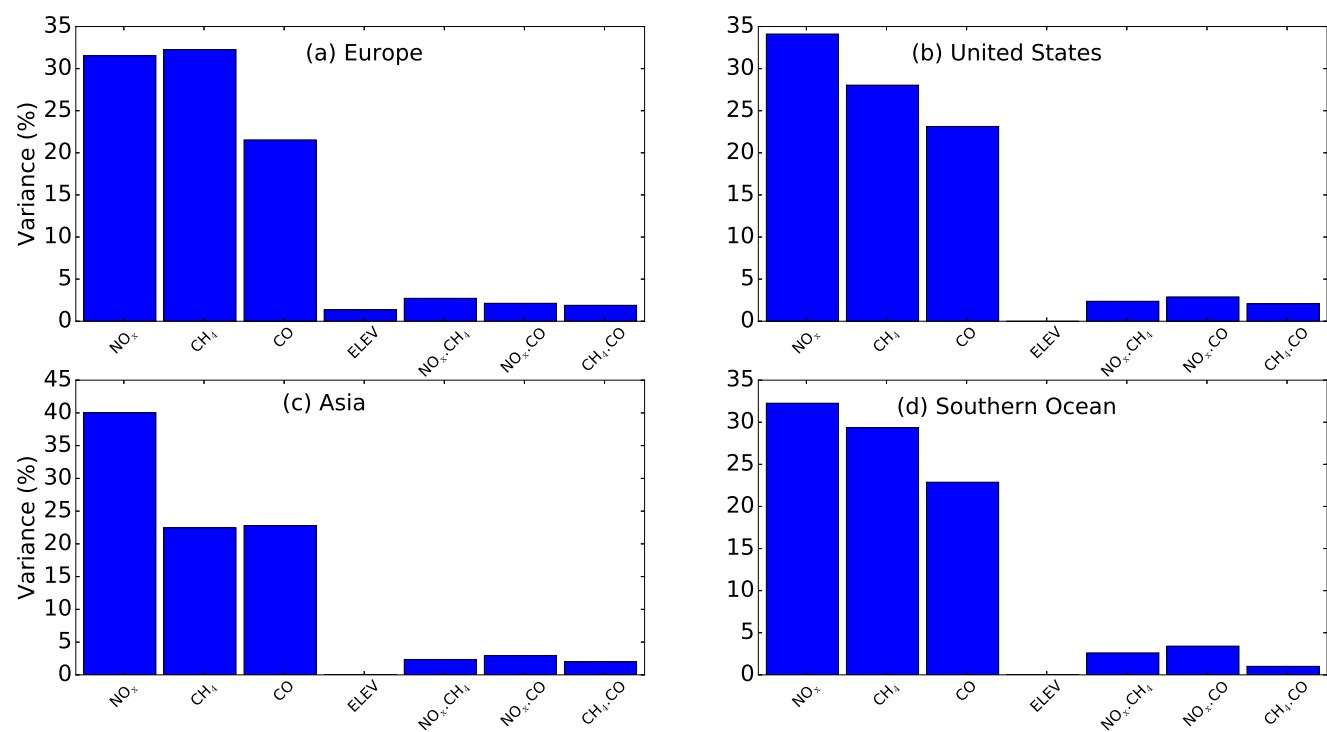

**Figure 8.** Contributions to variance from the 9 variables listed in Table 1. For clarity only those which contribute at least 1% are shown. $NO_x$ = $NO_x$ emissions; $CH_4$ = $CH_4$ concentrations; CO = CO+NMVOC emissions; ELEV = the number of vertical model levels that $NO_x$, CO and NMVOC emissions are prescribed on. Joint interactions, indicated by e.g. $NO_x.CH_4$ are also indicated where these contribute at least 1% to the variance.



**Table 1.** Minimum and maximum scalings applied to the sensitivity forcings/parametrizations. **P** and **L** indicate whether the variable is of relevance to ozone production and/or loss, respectively.

| | Minimum | Maximum | Comment |
|---|---|---|---|
| (1) $NO_x$ emissions **(P)** | 0 | 4 | The surface emissions field as a function of latitude and longitude was multiplied by the scaling factor. |
| (2) $CH_4$ concentrations **(P)** | 0 | 4 | The global-mean mixing ratio was multiplied by the scaling factor. |
| (3) CO+NMVOC emissions **(P)** | 0 | 4 | As for (1). |
| (4) ELEV for $NO_x$ and CO+NMVOCs **(P)** | 1 | 6 | Emissions were prescribed on the lowermost 1–6 levels (between the surface and $\sim$2.5 km). |
| (5) CLEV for $CH_4$ **(P)** | 1 | 6 | Concentrations were prescribed on the lowermost 1–6 levels (between the surface and $\sim$2.5 km). |
| (6) CMF **(P+L)** | 0.25 | 1 | 1 implies clear-sky photolysis, whereas 0 would imply no photolysis. |
| (7) $HNO_3$ washout **(L)** | 0 | 0.5 | Corresponds to removing between 0–50% of tropospheric gas-phase $HNO_3$ at each chemical time step. |
| (8) $N_2O_5$ hydrolysis **(L)** | 0.001 | 0.3 | The constant uptake coefficient (gamma); the default is 0.1. |
| (9) $O_3$ dry deposition **(L)** | 0 | 1 | A specific reactivity of 0 stands for a nearly non-reactive gas, while 1 stands for a gas similarly reactive to ozone. |