# Peer review of "Tropospheric ozone in CCMI models and Gaussian process emulation to understand biases in the SOCOLv3 chemistry-climate model"

_Atmospheric Chemistry and Physics, 2018_

## Referee Comment (RC1) · E. Ryan (Referee) · 1 Aug 2018

General comments

This is a nice paper. As a statistician, I focused mainly on the emulator and statistical part of the manuscript. So my first comment is that it's great to see emulators appearing in atmospheric chemistry modelling research for the purposes of doing statistical analyses such as global sensitivity analysis which would be too computationally burdensome without emulators. These papers are still fairly rare, so you're encouraging others in the atmospheric chemistry modelling community to consider these methods (which is awesome!). GEM-SA is a great tool developed by statisticians at the Univer-

sity of Sheffield, to make it easier for applied scientists to carry out this type of statistical analysis with minimal understanding of the statistics. The implementation of GEM-SA appears to be done correctly and so I'm satisfied that the results are all fine. However there are a large number of issues that need addressing, So although there is nothing major that needs changing, I've indicated major corrections to give you enough time to address these large number of comments, some of which made need a lot of thought. Feel free to e-mail me if you need me to clarify any of these comments.

Major Comments

[1] page 6, lines 20-21. The sentence starting "The output variable ..." sits uncomfortably with me. While we are technically "fitting", I would use this word here as the non-statistical reader may infer from this that you're using measurements. The phrase "uncertainties are calculated with a covariance function" is also too vague. Finally, you say that "each output point has a normal distribution." This is incorrect. A GP emulator that the guys at Sheffield developed is built within a Bayesian framework, where prior is a GP, the likelihood function is a multivariate Normal distribution and the resulting derived posterior is a student-t distribution. I suggest you drastically reword this sentence. You can still keep parts of it, but the parts above that I mention need to be changed. I suggest you use these few lines to actually define what a GP emulator is. In Tony O'Hagan's paper he defines it by two properties: (1) an interpolator such that at inputs the emulator is trained at, the emulated outputs must be the same as the simulator outputs; (2) for inputs the emulator is not trained at, the emulated outputs have a probability distribution specified by a mean function and a covariance function. In my paper that recently got accepted (Ryan et al., in review; https://www.geosci-model-dev-discuss.net/gmd-2017-271/), I give a definition like this and other details. You may want to refer to this to help with this part of your methods section.

[2] Page 7, line 24. I feel uncomfortable about you using the words "not necessarily feasible" here. For a sensitivity analysis study, justifying he mins and maxs of your inputs is important because if your range covers values of a particular input that are

not feasible this could give misleading results in the sensitivity analysis. In other words, suppose the range for an input is (2,4) and you find that the output is not sensitive to the changes in that input. Now suppose you were to repeat the analysis with a range of that input as (1,4) and suppose that the output is now quite sensitive to the changes in that input. Well, this means that the results of the sensitivity analysis are "sensitive" to the value you used for the minimum value. This won't always be the case, but I feel it's important to justify why the choices of mins and maxs of each input are appropriate.

[3] Page 8/9 (section 3.1). In your methods, I found only one line where you talk about incorporating other models in this study, but then in your results you have four figures (figs. 2-5) of results before getting onto the results from the sensitivity analysis. I am unsure how section 3.1 and figures 2-5 fit into this analysis. Please can you explain this? Have figures 2-5 been reported elsewhere? I can understand why you may want to include one or two of figures 2-5 in your methods and motivation for doing the sensitivity analysis, but I don't think they should be part of your results. Reading your abstract, it seems that your paper is split into two parts: (1) introducing a new version to the SOCOL model; (2) carrying out the sensitivity analysis. So I could understand if figures 2-5 and section 3.1 were devoted to validating or testing SOCOL v3.1, but including the other CCMI models in your "results" section seems problematic. If you do justify leaving in section 3.1 and figs 2-5 then at the very least I feel that you need to talk a lot more about these CCMI models in your methods and what research questions you're answering. Looking at the end of your introduction (where research questions are normally stated), the only things I read, that state what the paper will be about, are: (1) some results from SOCOL v3.1 and (2) the sensitivity analysis. Do you see my confusion?

Minor Comments

[1] In the abstract (page 2, lines 1-2), you talk about the reduction in ozone bias due to the inclusion of the N2O5 hydrolysis process. Is this reduction in bias at the cost of an increase in bias for other variables (e.g. CH4 lifetime) when compared with

observations? This isn't necessarily something you need to change in the abstract, but the inclusion of an extra sentence in the manuscript which addresses this comment would be useful.

[2] Page 2, line 6. "More than 90%"? Adding up the first three columns of figure 8, it looks more like 80-90%.

[3] In the title and elsewhere in the manuscript you mostly refer to the emulator as a "Gaussian emulator" (I found five mentions of this but there may be more). Please change all occurrences of this phrase to "Gaussian process emulator" (or "GP emulator" once GP is defined) since this is what you've implemented. A Gaussian (Normal) distribution is related to but is also quite different to a Gaussian process, so it's important to make this distinction. I'm guessing that you used 'Gaussian process' because of GEM-SA being short for 'Gaussian Emulation Machine for Sensitivity Analysis'. 'Gaussian emulation' was probably used here to make the acronym work, but it's unfortunately also caused confusion.

[4] Page 3, lines 20-26. You've got to be a bit careful about the language used here. You imply that it's the GP emulator that doing the Global Sensitivity Analysis (GSA). The point is that you need to do 1000s of runs to the GSA, so the emulator (trained with only 90 simulator runs) is much more computationally efficient. I know that you probably know this, but at the moment this isn't clear to me when I read these lines.

[5] Page 3, line 26. The word "non-linear" is the probably the wrong word to use here. I think what you're referring to are the sensitivity indices computed due the interaction of two inputs. If this is what you mean that I suggest you replace non-linear with "interacting".

[6] First line of section 2.4 (page 6). Please change the start of the sentence to "Variance-based global sensitivity analysis . . ."

[7] Page 3, line 30. Oliver Wild's group at Lancaster University are also using emulators

for their work with the FRSGC and GISS models. A paper of theirs which has been accepted and will be published shortly is (Ryan et al., 2018; https://www.geosci-model-dev-discuss.net/gmd-2017-271/). Please add the following to the end of this sentence on line 30: "... and to chemical transport modelling (Ryan et al., 2018)" or something to that effect.

[8] page 6, line 19 – what do you mean "supplement" here? Following the comma I suggest you replace the text with "..., a type of statistical model called Gaussian process emulator can be used as a surrogate for the input-output relation of the a complex model (Le Gratiet et al, 2017)." There are many other references from the statistics literature that could be included as well as the Le Gratiet ref.

[9] Page 7, 18. Can I suggest that you split this sentence beginning "90" into two sentences. The bit in brackets concerning the 10*n rule would be good to be taken out of the brackets and form the first sentence. Please also use the Loeppky et al. (2009) ref to justify the 10*n rule.

[10] Page 7, line 21. Replace "statistical method called" with "design" since this is what a Maximin LHD is.

[11] Page 7, line 22-23. On the line that follows, replace "approach" with "design". What do you mean by "near random sample"? This seems incorrect to me. Also the phrase "maximizing the uncertainty space" doesn't sit comfortably with me either. An Maximin LHD is a space filling design. It is an efficient design for sampling form a multi-dimensional parameter / input space in terms of being space filling but not requiring many samples. On page 169 of the pdf of my PhD thesis (given as page 155 in the footer) (Ryan et al., 2013), I give a fuller description if that'll help.

[12] Page 7, lines 21-23. How did you generate the Maximin LHD? I haven't used GEM-SA in a long time, so I can't remember if it has a feature which generates the design for you?

[13] Page 7, lines 21-24. I notice that some of your inputs are continuous (e.g. inputs 1-3) and some are discrete (e.g. input 4). Whenever I've built emulators, all of my inputs are continuous. Indeed, I think this is the norm when using a maximin LHD. For the statistical individuals like me reading this, please can you add in a line stating how you used this design for the inputs that are discrete? E.g. did you just round to the nearest whole number? Rounding to the nearest who number might be okay but it might not be. You might want to survey the literature a bit and what others have done.

[14] Page 9, line 26 – page 10, line 7. The first two paragraphs and start of the third paragraph of section 3.2 aren't anything to do with emulation or sensitivity analysis so please move to a different section or create a new section.

[15] page 10, line 9. Please don't use the word "correlation". Correlation is represented by 'r' and takes values between -1 and 1. R^2 is a measure of "goodness of fit" (takes values 0-1) which in this case refers to how well the emulated outputs compare with the simulator outputs at the validation inputs.

[16] Page 10, lines 17/18. You state here "... assuming all other parameters are held constant." This is wrong. This is what happens with one at a time sensitivity analysis. With variance based global sensitivity analysis, we average over the other inputs. See slide 9 of: https://view.officeapps.live.com/op/view.aspx?src=http://www.tonyohagan.co.uk/academic/GEM/SensitivityAnalysis.ppt

[17] Page 11, line 33. You mention Young et al. (2018). From memory this is one of the TOAR papers where the chemistry models are compared with observations from the newly formed TOAR network. If you are going to keep figs 2-5 in their current form, then it seems that Young et al. (2018) is a key paper that you need to refer to a lot earlier in the paper (e.g. intro and methods).

[18] Page 13. Data availability section. For the benefit of reproducibility, please can you make the matrix of inputs and outputs that were used in GEM-SA to generate your sensitivity analysis results.

[19] Figure 1: When we do variance based global sensitivity analysis, the inputs are normalized to all be between 0 and 1. I think GEM-SA does this automatically. I mention this because it would look a lot better if the y-axis on figure 1 referred to the normalized inputs. By normalized I mean: $x\_norm = (x - xmin)/(xmax-xmin)$. This would make the points in figure 1 appear more randomly scattered as opposed to the larger gaps for the higher values of the inputs because of only some of the inputs extend to 4 or 6.

[20] Figure 1. Are all of your inputs scaling factors? It seems not since for example input 4 is the "number of vertical levels …". If you agree, please change the y-axis label to "Inputs" or "parameters".

[21] Figure 6. The caption is quite short here. I know that in the methods you described the simulator runs for validation of the emulators as "test simulations". But at some point in this caption you need to explain that these runs correspond to running the emulators and simulators at each of the 27 validation inputs. You also need to explain what each of the panels refer to? I know it might seem obvious, but my view is that I should be able to understand everything about each figure without having to refer to the manuscript text. You also need to describe what $R^2$ is (not correlation, look it up on Wikipedia).

[22] Figure 7. In the caption please replace "assuming the other variables are constant" with "averaging over the other inputs." You state this plot is representative of the other regions. Please can you put the equivalent plots for the other regions in the supplemental material.

[23] Figure 8. Why not show the sensitivity indices for all nine inputs? I know that you say that you're not including the missing two because they are less than 1%, but for completeness (and given that it's only an extra two bars), I think it's worth including them.

[24] Table 1. Is it accurate to describe all the inputs has "scaling". E.g. input 4 is not a

scaling since it's the no. of level.s

[25] Table 1. You have a "Comments" column, but I think that replacing this with "Descriptions" and giving a full definition of what each input is would be better.

Reference

O'Hagan, A. (2006). Bayesian analysis of computer code outputs: A tutorial. Reliability Engineering & System Safety, 91(10-11), pp.1290-1300.

Loeppky, J. L., Sacks, J., & Welch, W. J. (2009). Choosing the sample size of a computer experiment: A practical guide. Technometrics, 51(4), 366-376.

Ryan, E. (2013). The Limitations and Robustness of Data Assimilation in Terrestrial Ecosystem Modelling (Doctoral dissertation, University of Sheffield). http://etheses.whiterose.ac.uk/4293/

Ryan, E., Wild, O., O'Connor, F., Voulgarakis, A., and Lee, L. (accepted, 2018): Fast sensitivity analysis methods for computationally expensive models with multidimensional output, Geosci. Model Dev. Discuss., https://doi.org/10.5194/gmd-2017-271.
* * *

---

## Referee Comment (RC2) · Anonymous Referee #2 · 25 Aug 2018

This manuscript quantifies tropospheric ozone biases in two versions of the SOCOL chemistry-climate model, as well as the CCMI models. The SOCOL bias is further investigated using an emulator. I find the methodology novel, and the Discussions and Conclusions is particularly well reasoned and should be of considerable interest to the chemistry-climate modeling community. I do believe the paper could be greatly improved if some choices and details of the methodology are better explained (and perhaps if the paper is slightly restructured) as I explain in my two major criticisms below.

General comments

[Figure]

1) A stronger rationalization of the input parameter choices for the emulator is needed in Section 2.4. An important reason for testing the ozone precursors [variables (1-3)] is that they are a primary candidate for the cause of the systematic high bias in tropospheric ozone among model intercomparisons that use harmonized emissions, such as CCMI and ACCMIP. An important reason for testing (3) should be that SOCOL is very simplistic in its representation of NMVOC chemistry compared to other CCMI models (as an aside: why not vary the yield of CO from NMVOC oxidation separately to the magnitude of NMVOC emissions?). It also seems that variables (4-9) are chosen to reflect developments between SOCOLv3.0 and v3.1...is that correct? If so, I am not sure why, besides (8), they are investigated at all since the authors have already performed a sensitivity test in which they find that inclusion of heterogeneous hydrolysis of N2O5 is the main development that reduces the model's ozone bias between the two versions (P9L31).

2) It seems that the most detailed portion of the paper is focused on quantifying and understanding SOCOL's ozone biases, in part with the emulator, rather than an exploration of biases in the CCMI models (which could be a paper by itself!). With this in mind, the authors might consider first discussing SOCOL biases and then placing the results of the single model study within the wider context of the CCMI models e.g. combining Section 3.1 with the first paragraph of the Discussions and Conclusions. However, I leave this up to the authors. Secondly, and more importantly, please elaborate upon the basics of the emulation technique. Although I appreciate that the authors are probably trying to avoid jargon, as a non-statistician, I find the beginning of Section 2.4 a little confusing. Finally, the emulator experiments are a novel contribution to this field, which should be emphasized in the Introduction and Conclusions to increase the significance of the paper. Perhaps the authors could also speak to the broader goals such as extending the emulation methodology to explore tropospheric ozone variability due to meteorological parameters (e.g. convective parameters) not investigated here, or variability in other metrics such as ozone extremes etc...

Specific comments

P2L21: these fractions were deduced using data over individual sites in the Southern Hemisphere and are not necessarily representative of the whole troposphere.

P2L23: specify that this is the "global tropospheric lifetime" since the ozone lifetime can vary considerably by region.

P2L27: please cite Young et al. (2018) alongside Young et al. (2013) and Parrish et al. (2014).

P3L5: please cite Stevenson et al. (2006) for ACCENT and Young et al. (2013) for ACCMIP.

P3L26 (and P6L21): Do you mean non-additive instead of non-linear?

P4L3: For clarity, specify that SOCOL is a chemistry-climate model.

P4: Provide some information about the stratospheric boundary conditions.

P4L16: A look-up table is an offline, not online, photolysis scheme (in agreement with the last sentence of the paragraph).

P5L14: This is inconsistent with P4L29, which states that methane is prescribed as a "surface mixing ratio", which implies the lowermost model level.

P5L16: Naively, I would not expect methane-induced ozone production to be reduced upon prescribing methane on one level versus multiple levels since it is well mixed in the troposphere.

P6 paragraph 1 and Section 3.1: I wonder how much of the inter-model differences in the tropospheric ozone burden arise from inter-model differences in tropopause height. Could this be quantified by imposing the same tropopause height across all the models and noting the difference in ozone burden?

P6L20: Please see General Comment #2. This sentence is packed with information

and is confusing to a non-statistician.

P6 points 1 and 3: Which type of emissions? Anthropogenic/biomass burning/natural?

P6 point 4: I am unclear as to why this is tested. Emissions are included as surface fluxes (i.e. lowest model level) in both SOCOL versions, and to my knowledge, across most models.

P7 point 5: I would have thought a priori that the number of levels that methane is prescribed on would not matter for tropospheric ozone amounts, and this is confirmed later in the paper.

P7L24: I am not sure why you would test ranges that are not feasible. E.g. the maximum range for methane (4xCH4) is much larger than even RCP8.5 year 2100 amounts relative to present day. Are we then sure the results of the emulator remain meaningful?

P7: The final paragraph explains that physical/meteorological parameters are, by design, not investigated in the emulator experiments. Indeed there could be multiple reasons, besides chemistry, for SOCOL's particularly high ozone bias. This is explained well in the Discussion, but should also be made clear in the Introduction: the methodology used here does not explain (nor is it intended to explain) the entirety of the "remaining ozone bias in SOCOLv3.1" as stated on P3L20.

P8L2 and Section 3.2: Why not also show results for the global mean tropospheric ozone burden, given its discussion in the Abstract and elsewhere.

P8L12: Reference Morgenstern et al. (2017) who discuss familial relationships between the CCMI models.

P8L22: I do not think you can say ECAM-L90 simulates a "better" representation here since there is no comparison to the observations yet.

P9L16: Please provide the ACCMIP MMM global mean tropospheric ozone burden in DU for comparison with CCMI and CMIP5. Also state which, or at least how many,

models were considered in the ACCMIP and CMIP mean.

P9: The CCMI/ACCMIP/CMIP5 comparison is brief. This is fine for the present study, but perhaps the authors could highlight the potential for more detailed future investigation (see also General Comment #2). It would be interesting to see the extent of agreement - or lack thereof - between the different model intercomparisons' simulation of tropospheric ozone, given their different aims and formulations (e.g. a focus on stratosphere-troposphere interactions in the CCMI models vs atmosphere-ocean coupling in CMIP5).

Figures 2 and parts of Figure 4, 5: The continuous scale in these figures makes it difficult to distinguish numerical differences between the sub-plots. I recommend a discrete scale as in Figures 3 and 4c, 4f, 5c, 5f.

P9L30: Do you mean regionally not globally?

P9L33: From Figure 3, it looks like several of the CCMI models also show this bias over the Southern Ocean. Do they share the Wesely deposition scheme?

P10L6: State where this maximum bias occurs.

P10L9, Figure 6: Am I right in thinking that two conditions need to be satisfied in order for the emulator to perform well: having a high R squared value and having the points falling on a 1:1 line? Please clarify.

P10L10: See earlier comment about using inputs outside feasible ranges, which is acknowledged on P10L30. Do these extremes need to be tested?

P10L20: Can we explain this? Does it reflect a NOx titration effect?

P10L17, Figure 7: I am a little confused on what to take from this figure: is the "sensitivity" of tropospheric ozone to each parameter determined by the slopes of the sub-plots? If so, why compare the different sensitivities? To determine which parameters are more "important" for tropospheric ozone variability, it makes more sense to compare the variance explained by each parameter (Figure 8). Finally, what does the uncertainty in Figure 7 signify? I may be missing the obvious! Please explain Figure 7 clearly or consider removing.

P10L17: "Figure 7 displays the sensitivity of global-mean tropospheric ozone..." but the figure caption suggests the mean is over the Asian region only.

Figure 8: Remove "9 variables" from the figure caption since all 9 variables are not shown.

Figure 8: Could you also show a panel for the global mean burden?

Figure 8: Could you explain why the relative importance of CH4 and CO is smaller over Asia than Europe or the US? It would be better to use the same scale on all the panels.

P11L6: "up to 8 DU regionally"

P11L12: "up to ∼30 DU regionally"

Discussions and Conclusions: I very much like this section! I would only conclude with some remarks on the novelty of the emulation technique within this field and its potential future value in the study of ozone biases (see General Comment #2).

References

Morgenstern, O., Hegglin, M. I., Rozanov, E., O'Connor, F. M., Abraham, N. L., Akiyoshi, H., Archibald, A. T., Bekki, S., Butchart, N., Chipperfield, M. P., Deushi, M., Dhomse, S. S., Garcia, R. R., Hardiman, S. C., Horowitz, L. W., Jöckel, P., Josse, B., Kinnison, D., Lin, M., Mancini, E., Manyin, M. E., Marchand, M., Marécal, V., Michou, M., Oman, L. D., Pitari, G., Plummer, D. A., Revell, L. E., Saint-Martin, D., Schofield, R., Stenke, A., Stone, K., Sudo, K., Tanaka, T. Y., Tilmes, S., Yamashita, Y., Yoshida, K., and Zeng, G.: Review of the global models used within phase 1 of the Chemistry–Climate Model Initiative (CCMI), Geosci. Model Dev., 10, 639-671, https://doi.org/10.5194/gmd-10-639-2017, 2017.

Stevenson, D. S., Dentener, F. J., Schultz, M. G., Ellingsen, K., van Noije, T. P. C., Wild, O., Zeng, G., Amann, M., Atherton, M., Bell, N., Bergmann, D. J., Bey, I., Bulter, T., Cofala, J., Collins, W. J., Derwent, R. G., Doherty, R. M., Drevet, J., Eskes, H. J., Fiore, A. M., Gauss, M., Hauglustaine, D. A., Horowitz, L. W., Isaksen, I. S. A., Krol, M. C., Lamarque, J.-F., Lawrence, M. G., Montanaro, V., Muller, J.-F., Pitari, G., Prather, M. J., Pyle, J. A., Rast, S., Rodriguez, J. M., Sanderson, M. G., Savage, N. H., Shindell, D. T., Strahan, S. E., Sudo, K., and Szopa, S.: Multimodel ensemble simulations of present-day and near-future tropospheric ozone, J. Geophys. Res., 111, D08301, doi:10.1029/2005JD006338, 2006.

Young, P. J., Archibald, A. T., Bowman, K. W., Lamarque, J.-F., Naik, V., Stevenson, D. S., Tilmes, S., Voulgarakis, A., Wild, O., Bergmann, D., Cameron-Smith, P., Cionni, I., Collins, W. J., Dalsøren, S. B., Doherty, R. M., Eyring, V., Faluvegi, G., Horowitz, L. W., Josse, B., Lee, Y. H., MacKenzie, I. A., Nagashima, T., Plummer, D. A., Righi, M., Rumbold, S. T., Skeie, R. B., Shindell, D. T., Strode, S. A., Sudo, K., Szopa, S., and Zeng, G.: Pre-industrial to end 21st century projections of tropospheric ozone from the Atmospheric Chemistry and Climate Model Intercomparison Project (ACCMIP), Atmos. Chem. Phys., 13, 2063-2090, https://doi.org/10.5194/acp-13-2063-2013, 2013.

Young P. J., Naik V., Fiore A. M., Gaudel A., Guo J., Lin M.Y., et al.. Tropospheric Ozone Assessment Report: Assessment of global-scale model performance for global and regional ozone distributions, variability, and trends. Elem Sci Anth. 2018;6(1):10. DOI: http://doi.org/10.1525/elementa.265
* * *

---

## Author Comment (AC1) · 12 Sep 2018

E. Ryan (Referee)

edmund.ryan@lancaster.ac.uk

General comments

This is a nice paper. As a statistician, I focused mainly on the emulator and statistical part of the manuscript. So my first comment is that it's great to see emulators appearing in atmospheric chemistry modelling research for the purposes of doing statistical analyses such as global sensitivity analysis which would be too computationally burdensome without emulators. These papers are still fairly rare, so you're encouraging others in the atmospheric chemistry modelling community to consider these methods (which is awesome!). GEM-SA is a great tool developed by statisticians at the University of Sheffield, to make it easier for applied scientists to carry out this type of statistical analysis with minimal understanding of the statistics. The implementation of GEM-SA appears to be done correctly and so I'm satisfied that the results are all fine. However there are a large number of issues that need addressing, So although there is nothing major that needs changing, I've indicated major corrections to give you enough time to address these large number of comments, some of which made need a lot of thought. Feel free to e-mail me if you need me to clarify any of these comments.

**Major Comments**

[1] page 6, lines 20-21. The sentence starting "The output variable . . ." sits uncomfortably with me. While we are technically "fitting", I would use this word here as the non-statistical reader may infer from this that you're using measurements. The phrase "uncertainties are calculated with a covariance function" is also too vague. Finally, you say that "each output point has a normal distribution." This is incorrect. A GP emulator that the guys at Sheffield

developed is built within a Bayesian framework, where prior is a GP, the likelihood function is a multivariate Normal distribution and the resulting derived posterior is a student-t distribution. I suggest you drastically reword this sentence. You can still keep parts of it, but the parts above that I mention need to be changed. I suggest you use these few lines to actually define what a GP emulator is. In Tony O'Hagan's paper he defines it by two properties: (1) an interpolator such that at inputs the emulator is trained at, the emulated outputs must be the same as the simulator outputs; (2) for inputs the emulator is not trained at, the emulated outputs have a probability distribution specified by a mean function and a covariance function. In my paper that recently got accepted (Ryan et al., in review; https://www.geosci-modeldev-discuss.net/gmd-2017-271/), I give a definition like this and other details. You may want to refer to this to help with this part of your methods section.

This section has been rewritten:

"Variance-based global sensitivity analysis allows the individual contribution of a single parameter to the overall uncertainty to be quantified. Because the large number of model simulations required would make one-at-a-time testing computationally too expensive, a type of statistical model called a GP emulator can be used as a surrogate for the input-output relation of a complex model, such as a CCM (Le Gratiet et al., 2017). For "training" data on which the GP emulator is built, we know that the true value of the emulated output should be the same as the input, so the emulator should return the output with no uncertainty. For inputs that the emulator is not trained at, the outputs should have a probability distribution specified by a mean function and covariance function (O'Hagan, 2006). Here, we use tropospheric ozone columns from SOCOLv3.1 to train the emulator.

[2] Page 7, line 24. I feel uncomfortable about you using the words "not necessarily feasible" here. For a sensitivity analysis study, justifying he mins and maxs of your inputs is important because if your range covers values of a particular input that are not feasible this could give misleading results in the sensitivity analysis. In other words, suppose the range for an input is (2,4) and you find that the output is not sensitive to the changes in that input. Now suppose you were to repeat the analysis with a range of that input as (1,4) and suppose that the output is now quite sensitive to the changes in that input. Well, this means that the results of the sensitivity analysis are "sensitive" to the value you used for the minimum value. This won't always be the case, but I feel it's important to justify why the choices of mins and maxs of each input are appropriate.

We have removed this sentence, and expanded Table 1 to add further description of the ranges for the sensitivity analysis. The revised Table 1 is shown below. Some of the ranges were chosen based on past experience with SOCOL – for example, previous sensitivity tests have indicated that halving the $NO_x$ emissions leads to close agreement between modelled and observed tropospheric column ozone. Here the range of 0.25 to 4 was selected to cover a larger uncertainty space. For ELEV and CLEV, the maximum of 6 levels (~2.5 km) corresponds to the maximum boundary layer height at mid-latitudes, which is where most

emissions occur; however most (if not all) models prescribe emissions only at the surface, which is the recommended approach.

**Table 1.** Range of the sensitivity forcings/parametrizations. **P** and **L** indicate whether the variable is of relevance to ozone production and/or loss, respectively.

| | Minimum | Maximum | Descriptions |
|---|---|---|---|
| (1) $NO_x$ emissions (**P**) | 0 | 4 | The surface $NO_x$ emissions field as a function of latitude and longitude was multiplied by a scaling factor between 0 and 4, to explore the sensitivity of tropospheric ozone to a range of $NO_x$ emissions. |
| (2) $CH_4$ concentrations (**P**) | 0 | 4 | The global-mean $CH_4$ mixing ratio was multiplied by a scaling factor between 0 and 4, to explore the sensitivity of tropospheric ozone to a range of $CH_4$ concentrations. |
| (3) CO+NMVOC (**P**) emissions | 0 | 4 | As for (1), but the scaling factor was applied to CO and NMVOC emissions simultaneously. |
| (4) ELEV for $NO_x$ and CO+NMVOCs (**P**) | 1 | 6 | Emissions were prescribed on the lowermost 1–6 levels (between (the surface and ∼2.5 km, to test whether the number of levels is important for tropospheric ozone abundances. |
| (5) CLEV for $CH_4$ (**P**) | 1 | 6 | $CH_4$ concentrations were prescribed on the lowermost 1–6 levels (between the surface and ∼2.5 km, similar to (4). |
| (6) CMF (**P+L**) | 0.25 | 1 | 1 implies clear-sky photolysis, whereas 0 would imply no photolysis. As photolysis rates of 0 do not occur during daytime, we selected a lower bound of 0.25 to represent cloudy sky conditions. |
| (7) $HNO_3$ washout (**L**) | 0 | 0.5 | To test the sensitivity of tropospheric ozone to $HNO_3$ removal, we removed between 0–50% of tropospheric gas-phase $HNO_3$ at each chemical time step. |
| (8) $N_2O_5$ hydrolysis (**L**) | 0.001 | 0.3 | The probability of $N_2O_5$ hydrolysis occurring. Since the default is 0.1, we explored the sensitivity of tropospheric ozone to a range from 0.001-0.3. |
| (9) $O_3$ dry deposition (**L**) | 0 | 1 | A specific reactivity of 0 stands for a nearly non-reactive gas, while 1 stands for a gas similarly reactive to ozone. |

[3] Page 8/9 (section 3.1). In your methods, I found only one line where you talk about incorporating other models in this study, but then in your results you have four figures (figs. 2-5) of results before getting onto the results from the sensitivity analysis. I am unsure how section 3.1 and figures 2-5 fit into this analysis. Please can you explain this? Have figures 2-5 been reported elsewhere? I can understand why you may want to include one or two of figures 2-5 in your methods and motivation for doing the sensitivity analysis, but I don't think they should be part of your results. Reading your abstract, it seems that your paper is split into two parts: (1) introducing a new version to the SOCOL model; (2) carrying out the sensitivity analysis. So I could understand if figures 2-5 and section 3.1 were devoted to validating or testing SOCOL v3.1, but including the other CCMI models in your "results" section seems problematic. If you do justify leaving in section 3.1 and figs 2-5 then at the very

least I feel that you need to talk a lot more about these CCMI models in your methods and what research questions you're answering. Looking at the end of your introduction (where research questions are normally stated), the only things I read, that state what the paper will be about, are: (1) some results from SOCOL v3.1 and (2) the sensitivity analysis. Do you see my confusion?

The CCMI aspect of the study is an important one, as this is the first time that global distributions of tropospheric ozone from the CCMI models have been presented and compared with observations. Following Reviewer 2's suggestion, we have shuffled the order of material in the Results and Discussion a little, so that the emulator results are presented before the comparison of the CCMI models.

In the revised manuscript, the CCMI comparison is described in:

- Abstract, lines 2-6:
    "We investigate annual-mean tropospheric column ozone in 15 models participating in the SPARC/IGAC (Stratosphere-troposphere Processes and their Role in Climate/International Global Atmospheric Chemistry) Chemistry-Climate Model Initiative (CCMI). These models exhibit a positive bias, on average, of up to 40–50% in the Northern Hemisphere compared with observations derived from the Ozone Monitoring Instrument and Microwave Limb Sounder (OMI/MLS), and a negative bias of up to ~30% in the Southern Hemisphere."
- Introduction, P3L30-P4L2:
    "SOCOLv3.0 participated in phase 1 of the Chemistry-Climate Model Initiative (CCMI) (Eyring et al., 2013; Morgenstern et al., 2017), which is a joint activity of SPARC (Stratosphere-troposphere processes and their role in Climate) and IGAC (International Global Atmospheric Chemistry), and is the successor activity to phase 2 of the Chemistry-Climate Model Validation activity, CCMVal-2 (SPARC CCMVal, 2010). Unlike CCMVal-2, which focussed on stratospheric processes and composition, CCMI includes many models with comprehensive representations of the troposphere, and aims to additionally address aspects of tropospheric chemistry and circulation. Here, we examine tropospheric column ozone in SOCOLv3.0 and 14 other CCMI models. This is the first time that global distributions of tropospheric ozone have been examined in the CCMI models, and results are presented in Section 3.3."
- Methods, section 2.1 ("CCM simulations to compare with observations.")

**Minor Comments**

[1] In the abstract (page 2, lines 1-2), you talk about the reduction in ozone bias due to the inclusion of the N2O5 hydrolysis process. Is this reduction in bias at the cost of an increase in bias for other variables (e.g. CH4 lifetime) when compared with observations? This isn't necessarily something you need to change in the abstract, but the inclusion of an extra sentence in the manuscript which addresses this comment would be useful.

If anything, calculated quantities such as the $CH_4$ lifetime should improve due to reductions in OH abundances ($CH_4$ + OH being the primary CH4 oxidation reaction). Historically SOCOLv3's simulated OH abundance has been too high, since ozone is the primary source of OH. Revell et al. (2015, www.atmos-chem-phys.net/15/5887/2015/) showed that this leads

to approximately 40 ppbv too little CO in the Northern Hemisphere compared with observations, because too much OH means too much CO is oxidised by CO + OH. Similarly, SOCOLv3's $CH_4$ lifetime was historically shorter than that calculated by other models. While the appropriate chemical reactions to calculate the $CH_4$ lifetime were not saved from our simulations, the simulated CO abundance has improved (the bias of -40 ppbv c.f. observations shown by Revell et al. (2015) has weakened to only -20 ppbv), and we have included a paragraph on that in the Discussion:

"Reducing SOCOL's tropospheric ozone bias is expected to lead to improvements in the simulated abundance of species which are oxidised by the hydroxyl radical, such as CO and $CH_4$, since ozone is the primary source of OH. Revell et al. (2015) showed that CO in SOCOLv3 was up to 40 ppbv too low in the Northern Hemisphere compared with observations from TES, due to the tropospheric ozone bias. In SOCOLv3.1, the Northern Hemisphere CO bias is reduced by approximately a factor of 2 (not shown)."

[2] Page 2, line 6. "More than 90%"? Adding up the first three columns of figure 8, it looks more like 80-90%.

When the joint interaction terms (NOx.CH4, NOx.CO and CH4.CO) are included, it comes to over 90% for all regions shown in Figure 8 (now Figure 5).

[3] In the title and elsewhere in the manuscript you mostly refer to the emulator as a "Gaussian emulator" (I found five mentions of this but there may be more). Please change all occurrences of this phrase to "Gaussian process emulator" (or "GP emulator" once GP is defined) since this is what you've implemented. A Gaussian (Normal) distribution is related to but is also quite different to a Gaussian process, so it's important to make this distinction. I'm guessing that you used 'Gaussian process' because of GEM-SA being short for 'Gaussian Emulation Machine for Sensitivity Analysis'. 'Gaussian emulation' was probably used here to make the acronym work, but it's unfortunately also caused confusion.

Thanks for explaining this! It has been changed to GP emulator throughout the manuscript.

[4] Page 3, lines 20-26. You've got to be a bit careful about the language used here. You imply that it's the GP emulator that doing the Global Sensitivity Analysis (GSA). The point is that you need to do 1000s of runs to the GSA, so the emulator (trained with only 90 simulator runs) is much more computationally efficient. I know that you probably know this, but at the moment this isn't clear to me when I read these lines.

This has been re-worded:

"Because thousands of simulations are required to perform a sensitivity analysis, and this would be computationally inefficient with a CCM, we supplement SOCOLv3.1 with a GP emulator. This allows a sensitivity analysis to be performed at low computational cost. Variance-based sensitivity analysis evaluates a suite of model input parameters, and their relationship to the variable of interest, simultaneously."

[5] Page 3, line 26. The word "non-linear" is the probably the wrong word to use here. I think what you're referring to are the sensitivity indices computed due the interaction of two inputs. If this is what you mean that I suggest you replace non-linear with "interacting".

Replaced as suggested.

[6] First line of section 2.4 (page 6). Please change the start of the sentence to "Variance-based global sensitivity analysis . . ."

Replaced as suggested.

[7] Page 3, line 30. Oliver Wild's group at Lancaster University are also using emulators for their work with the FRSGC and GISS models. A paper of theirs which has been accepted and will be published shortly is (Ryan et al., 2018; https://www.geosci-modeldev-discuss.net/gmd-2017-271/). Please add the following to the end of this sentence on line 30: ". . . and to chemical transport modelling (Ryan et al., 2018)" or something to that effect.

Done and thanks for the pointer to your paper.

[8] page 6, line 19 – what do you mean "supplement" here? Following the comma I suggest you replace the text with "..., a type of statistical model called Gaussian process emulator can be used as a surrogate for the input-output relation of the a complex model (Le Gratiet et al, 2017)." There are many other references from the statistics literature that could be included as well as the Le Gratiet ref.

Replaced as suggested.

[9] Page 7, 18. Can I suggest that you split this sentence beginning "90" into two sentences. The bit in brackets concerning the 10*n rule would be good to be taken out of the brackets and form the first sentence. Please also use the Loeppky et al. (2009) ref to justify the 10*n rule.

Replaced as suggested:

"Typically $10n$ simulations are recommended for training a GP emulator, where $n$ is the number of parameters under investigation (Loeppky et al., 2009). Hence we performed 90 SOCOLv3.1 "training'" simulations, and used the resulting annual-mean tropospheric ozone column to construct the GP emulator in several geographical regions (Europe, United States, Asia, the Southern Ocean and the global mean).

[10] Page 7, line 21. Replace "statistical method called" with "design" since this is what a Maximin LHD is.

Replaced as suggested.

[11] Page 7, line 22-23. On the line that follows, replace "approach" with "design". What do you mean by "near random sample"? This seems incorrect to me. Also the phrase "maximizing the uncertainty space" doesn't sit comfortably with me either. A Maximin LHD

is a space filling design. It is an efficient design for sampling form a multidimensional parameter / input space in terms of being space filling but not requiring many samples. On page 169 of the pdf of my PhD thesis (given as page 155 in the footer) (Ryan et al., 2013), I give a fuller description if that'll help.

This sentence has been changed:

"For each of the 90 training simulations, the 9 input variables were scaled simultaneously, with the scaling factors determined using a "maximin" Latin hypercube design, which generates a random sample of parameter values from a multidimensional distribution and fills the uncertainty space of the parameters (McKay et al., 1979)."

[12] Page 7, lines 21-23. How did you generate the Maximin LHD? I haven't used GEM-SA in a long time, so I can't remember if it has a feature which generates the design for you?

Yes, GEM-SA can generate Latin hypercubes, and this is what was done here. This has now been noted in the text.

[13] Page 7, lines 21-24. I notice that some of your inputs are continuous (e.g. inputs 1-3) and some are discrete (e.g. input 4). Whenever I've built emulators, all of my inputs are continuous. Indeed, I think this is the norm when using a maximin LHD. For the statistical individuals like me reading this, please can you add in a line stating how you used this design for the inputs that are discrete? E.g. did you just round to the nearest whole number? Rounding to the nearest who number might be okay but it might not be. You might want to survey the literature a bit and what others have done.

Added: "The Latin hypercube was generated using GEM-SA. For the discrete input parameters (e.g. (4) and (5) in the list above), the scaling factor was rounded to the nearest whole number."

[14] Page 9, line 26 – page 10, line 7. The first two paragraphs and start of the third paragraph of section 3.2 aren't anything to do with emulation or sensitivity analysis so please move to a different section or create a new section.

Created a new section, "Tropospheric ozone in SOCOLv3.1."

[15] page 10, line 9. Please don't use the word "correlation". Correlation is represented by 'r' and takes values between -1 and 1. R^2 is a measure of "goodness of fit" (takes values 0-1) which in this case refers to how well the emulated outputs compare with the simulator outputs at the validation inputs.

This has been corrected.

[16] Page 10, lines 17/18. You state here ". . . assuming all other parameters are held constant." This is wrong. This is what happens with one at a time sensitivity analysis. With variance based global sensitivity analysis, we average over the other inputs. See slide 9 of:

https://view.officeapps.live.com/op/view.aspx?src=http://www.tonyohagan.co.uk/academic/GEM/SensitivityAnalysis.ppt

This has been corrected.

[17] Page 11, line 33. You mention Young et al. (2018). From memory this is one of the TOAR papers where the chemistry models are compared with observations from the newly formed TOAR network. If you are going to keep figs 2-5 in their current form, then it seems that Young et al. (2018) is a key paper that you need to refer to a lot earlier in the paper (e.g. intro and methods).

This is now cited in the Introduction, as also requested by Reviewer 2:

"Most chemistry-climate models (CCMs), which are used to understand chemistry-climate interactions and project future atmospheric composition, overestimate tropospheric ozone in the Northern Hemisphere compared with observations (Young et al., 2013; Parrish et al., 2014; Young et al., 2018)."

[18] Page 13. Data availability section. For the benefit of reproducibility, please can you make the matrix of inputs and outputs that were used in GEM-SA to generate your sensitivity analysis results.

Certainly; these are now available in the supplement.

[19] Figure 1: When we do variance based global sensitivity analysis, the inputs are normalized to all be between 0 and 1. I think GEM-SA does this automatically. I mention this because it would look a lot better if the y-axis on figure 1 referred to the normalized inputs. By normalized I mean: x_norm = (x – xmin)/(xmax-xmin). This would make the points in figure 1 appear more randomly scattered as opposed to the larger gaps for the higher values of the inputs because of only some of the inputs extend to 4 or 6.

This has been changed as suggested.

[20] Figure 1. Are all of your inputs scaling factors? It seems not since for example input 4 is the "number of vertical levels . . .". If you agree, please change the y-axis label to "Inputs" or "parameters".

Changed to "Inputs."

[21] Figure 6. The caption is quite short here. I know that in the methods you described the simulator runs for validation of the emulators as "test simulations". But at some point in this caption you need to explain that these runs correspond to running the emulators and simulators at each of the 27 validation inputs. You also need to explain what each of the

panels refer to? I know it might seem obvious, but my view is that I should be able to understand everything about each figure without having to refer to the manuscript text. You also need to describe what R^2 is (not correlation, look it up on Wikipedia).

The caption has been changed to:

"Tropospheric column ozone as predicted by the GP emulator, vs. the amount simulated in SOCOLv3.1 "test" simulations (i.e., the simulations used to validate the emulator). The errorbars indicate the uncertainty on the GP emulator output, and the 1:1 line and coefficient of determination ($R^2$ value) are also shown. These simulations correspond to running the GP emulator and the simulator (SOCOLv3.1) at each of the 27 validation inputs, for: (a) Europe (37-60° N), 0-42° E); (b) United States (32-52° N, 67-124° W); (c) Asia (6-49° N, 70-146° E), (d) the Southern Ocean (45–60° S, all longitudes); and (e) globally."

[22] Figure 7. In the caption please replace "assuming the other variables are constant" with "averaging over the other inputs." You state this plot is representative of the other regions. Please can you put the equivalent plots for the other regions in the supplemental material.

Changed as suggested, and the equivalent plots are now in the supplement.

[23] Figure 8. Why not show the sensitivity indices for all nine inputs? I know that you say that you're not including the missing two because they are less than 1%, but for completeness (and given that it's only an extra two bars), I think it's worth including them.

We also show the joint interaction terms (e.g. NOx.CH4), making 45 possible terms to show in total – hence the decision to limit the number of terms plotted.

[24] Table 1. Is it accurate to describe all the inputs has "scaling". E.g. input 4 is not a scaling since it's the no. of levels.

Good point – it has been re-labeled as "range of the sensitivity forcings/parametrizations."

[25] Table 1. You have a "Comments" column, but I think that replacing this with "Descriptions" and giving a full definition of what each input is would be better.

Done, as suggested. The revised table is shown above.

---

## Author Comment (AC2) · 12 Sep 2018

This manuscript quantifies tropospheric ozone biases in two versions of the SOCOL chemistry-climate model, as well as the CCMI models. The SOCOL bias is further investigated using an emulator. I find the methodology novel, and the Discussions and Conclusions is particularly well reasoned and should be of considerable interest to the chemistry-climate modeling community. I do believe the paper could be greatly improved if some choices and details of the methodology are better explained (and perhaps if the paper is slightly restructured) as I explain in my two major criticisms below.

General comments

1) A stronger rationalization of the input parameter choices for the emulator is needed in Section 2.4. An important reason for testing the ozone precursors [variables (1- 3)] is that they are a primary candidate for the cause of the systematic high bias in tropospheric ozone among model intercomparisons that use harmonized emissions, such as CCMI and ACCMIP. An important reason for testing (3) should be that SOCOL is very simplistic in its representation of NMVOC chemistry compared to other CCMI models (as an aside: why not vary the yield of CO from NMVOC oxidation separately to the magnitude of NMVOC emissions?). It also seems that variables (4-9) are chosen to reflect developments between SOCOLv3.0 and v3.1...is that correct? If so, I am not sure why, besides (8), they are investigated at all since the authors have already performed a sensitivity test in which they find that inclusion of heterogeneous hydrolysis of N2O5 is the main development that reduces the model's ozone bias between the two versions (P9L31).

This section has been rewritten, taking the reviewers' feedback into account. To briefly answer the questions above:

a) SOCOL's NMVOC chemistry scheme is indeed very simplistic, and CO emissions far exceed NMVOC emissions (isoprene and formaldehyde). As described in the methodology, CO is prescribed as an "additional" NMVOC in SOCOL to account for missing NMVOCs. It is for this reason that we decided to treat CO and NMVOCs together. However, as we have now noted in the manuscript (see below), we do not recommend such an approach for CCMs with more complex NMVOC schemes.

b) Yes, variables 4-9 were chosen to reflect developments between SOCOLv3.0 and 3.1 – hopefully this is now clear in the revised text (below). Even though we performed some individual sensitivity tests initially, the advantage of including them in the sensitivity analysis is that joint interactions between these variables can be identified.

Revised text in Section 2.4 is as follows:

"Although many factors influence the tropospheric ozone budget, we restricted our analysis to 9 model forcings/parametrizations (see Table 1 for details of the scalings applied). These are listed below, followed by a section rationalizing the inclusion of each variable. We reiterate that this list above does not constitute a comprehensive list of variables controlling tropospheric ozone, however by illustrating the methodology used, we aim to demonstrate its utility."

…[List follows here]…

"Variables (1-3) were selected due to their importance as tropospheric ozone precursors. CO and NMVOC emissions were varied simultaneously (3) because the only NMVOCs included explicitly in SOCOL are isoprene and formaldehyde; other NMVOCs are represented via additional CO using a 'lumped' approach (Section 2.2). For models with a more complex representation of NMVOCs, we recommend testing CO and NMVOC emissions separately when constructing a GP emulator.

The remaining variables were included to investigate the sensitivity of tropospheric ozone to the model improvements implemented in SOCOLv3.1. SOCOLv3.0 and its predecessors prescribed methane on the lowermost six model levels. This was changed to only the surface level in SOCOLv3.1, and variable (5) was included in our analysis to investigate the sensitivity of tropospheric ozone to this implementation. By doing so, we aim to test the exchange of emissions between the boundary layer and free troposphere. The lowermost level in SOCOL covers approximately 100 m, and the 6 lowermost levels combined cover approximately 2.5 km. To explore whether other ozone precursors are sensitive to the number of levels they are prescribed on, variable (4) was included, even though it is prescribed only as a surface emissions flux in most, if not all, CCMs.

Because ozone production and destruction reactions are mostly photochemical, i.e. they occur in the presence of sunlight, we selected variable (6) to test the sensitivity of the current CMF parametrization, and examine impacts of the updated LUTs on tropospheric ozone in SOCOLv3.1. $HNO_3$ washout is the main sink for $NO_x$, and therefore affects the ozone budget. Future SOCOL versions will include an online wet deposition scheme, and so variable (7) was

selected to probe the sensitivity of tropospheric ozone to the rate of $HNO_3$ loss. Heterogeneous $N_2O_5$ hydrolysis is similarly important as it leads to $HNO_3$ formation, however it was not included in SOCOLv3.0. Therefore variable (8) was included in our analysis to quantify its relevance for tropospheric ozone abundances. Finally, variable (9) was chosen to test the sensitivity of tropospheric ozone to the newly-implemented dry deposition parametrization (Section 2.3)."

2) It seems that the most detailed portion of the paper is focused on quantifying and understanding SOCOL's ozone biases, in part with the emulator, rather than an exploration of biases in the CCMI models (which could be a paper by itself!). With this in mind, the authors might consider first discussing SOCOL biases and then placing the results of the single model study within the wider context of the CCMI models e.g. combining Section 3.1 with the first paragraph of the Discussions and Conclusions. However, I leave this up to the authors.

We have taken this suggestion on board, and shuffled material around; the methods subsection "CCM simulations to compare with observations" has been moved to the start of the methods section, and the results subsection "Tropospheric ozone in the CCMI models" has been moved to the end of the results section. This allows a more-or-less seamless transition from: a) describing the GP emulator methodology to showing the emulator results; and b) showing the CCMI comparison to discussing the results in the Discussion and conclusions.

Secondly, and more importantly, please elaborate upon the basics of the emulation technique. Although I appreciate that the authors are probably trying to avoid jargon, as a non-statistician, I find the beginning of Section 2.4 a little confusing.

Here we refer also to Referee 1's comments and our response to those. Referee 1 has previous experience with Gaussian Process emulation, and provided many constructive comments aimed at improving the description of this technique. We hope that the revised manuscript is now clearer to read for statisticians and non-statisticians alike.

Finally, the emulator experiments are a novel contribution to this field, which should be emphasized in the Introduction and Conclusions to increase the significance of the paper. Perhaps the authors could also speak to the broader goals such as extending the emulation methodology to explore tropospheric ozone variability due to meteorological parameters (e.g. convective parameters) not investigated here, or variability in other metrics such as ozone extremes etc...

We have emphasized the novel contribution of this study in the introduction and conclusions as suggested. E.g., from the Introduction:

"This is the first time the technique has been applied to global tropospheric ozone. Our GP emulator experiments have been designed to focus on recent developments regarding

SOCOL's tropospheric chemistry scheme, however the methodology has the potential to be expanded to also include meteorological parameters."

And the end of the Discussion and conclusions section:

"Given the results of our multi-model intercomparison as well as previous multi-model studies, our results highlight the need for careful validation of emissions inventories used by global models. However, the way in which emissions are handled by the models also appears to result in biased ozone abundances, and further work is needed to address the challenges of simulating sub-grid processes of importance to tropospheric ozone, in SOCOLv3 as well as in other CCMs. GP emulation may prove a useful tool for such studies, and we have demonstrated its usefulness for understanding tropospheric ozone biases. GP emulation is a powerful tool, and should be considered for use by those wanting to perform detailed sensitivity analyses at low computational cost."

Specific comments

3) P2L21: these fractions were deduced using data over individual sites in the Southern Hemisphere and are not necessarily representative of the whole troposphere.

Noted: "Greenslade et al. 2017 calculate the mean fraction of total tropospheric ozone attributable to STE at three sites between 38—69° S as 1-3%, and show that during individual STE events, over 10% of tropospheric ozone may be directly transported from the stratosphere."

4) P2L23: specify that this is the "global tropospheric lifetime" since the ozone lifetime can vary considerably by region.

Changed as suggested.

5) P2L27: please cite Young et al. (2018) alongside Young et al. (2013) and Parrish et al. (2014).

Done.

6) P3L5: please cite Stevenson et al. (2006) for ACCENT and Young et al. (2013) for ACCMIP.

Done.

7) P3L26 (and P6L21): Do you mean non-additive instead of non-linear?

Indeed, it turns out that non-linear is not the correct term – the other reviewer advised referring to them as "interacting" contributions, which we have now done.

8) P4L3: For clarity, specify that SOCOL is a chemistry-climate model.

Done.

9) P4: Provide some information about the stratospheric boundary conditions.

This information has been added to the section "CCM simulations to compare with observations." (Added text is in bold):

"Greenhouse gas concentrations ($CH_4$, $CO_2$ and $N_2O$) follow observations until 2005, then Representative Concentration Pathway (RCP) 8.5 (Riahi et al., 2011). Ozone precursor emissions (including $NO_x$, CO and NMVOCs) follow a historical emissions inventory until 2000 (Lamarque et al., 2010), then RCP 6.0 (Masui et al., 2011). Sea surface temperatures and sea ice concentrations were prescribed following HadISST observations (Rayner et al., 2003). **Concentrations of ozone-depleting substances followed the World Meteorological Organization's A1 scenario (WMO2011), and stratospheric aerosol surface area densities and optical parameters were prescribed from the SAGE-4λ data set (Arfeuille et al. 2013, Luo 2013)."**

10) P4L16: A look-up table is an offline, not online, photolysis scheme (in agreement with the last sentence of the paragraph).

This has been corrected.

11) P5L14: This is inconsistent with P4L29, which states that methane is prescribed as a "surface mixing ratio", which implies the lowermost model level.

That sentence has now been removed from P4L29, and the discussion about how methane is prescribed is left until the section "Upgraded model version SOCOLv3.1".

12) P5L16: Naively, I would not expect methane-induced ozone production to be reduced upon prescribing methane on one level versus multiple levels since it is well mixed in the troposphere.

We in the SOCOL group were also initially surprised at the result, however the reduction in tropospheric ozone is not huge (10% at maximum). Our reasoning for the result is outlined in the next few lines of the ACPD manuscript.

13) P6 paragraph 1 and Section 3.1: I wonder how much of the inter-model differences in the tropospheric ozone burden arise from inter-model differences in tropopause height. Could this be quantified by imposing the same tropopause height across all the models and noting the difference in ozone burden?

Shown below is annual-mean tropospheric column ozone in 2005, where tropospheric ozone columns were calculated between the surface and 250 hPa, rather than the WMO-defined tropopause. As would be expected by imposing the tropopause at 250 hPa, the global-mean tropospheric ozone abundance is smaller compared with Figure 2 from our ACPD manuscript. It can also be seen that the same differences in terms of the spatial distribution and tropospheric ozone abundances exist, regardless of where the tropopause is defined. As noted in the manuscript, we opted to select the WMO-defined tropopause to enable a "likewith-like" comparison with the OMI/MLS satellite product. Therefore the figure shown in the manuscript remains unchanged.

[Figure]

14) P6L20: Please see General Comment #2. This sentence is packed with information and is confusing to a non-statistician.

This section now reads:

"Variance-based global sensitivity analysis allows the individual contribution of a single parameter to the overall uncertainty to be quantified. Because the large number of model simulations required would make one-at-a-time testing computationally too expensive, a type of statistical model called a GP emulator can be used as a surrogate for the input-output relation of a complex model (Le Gratiet et al., 2017), such as a CCM. For "training" data on which the GP emulator is built, we know that the true value of the emulated output should be the same as the input, so the emulator should return the output with no uncertainty. For inputs that the emulator is not trained at, the outputs should have a probability distribution specified by a mean function and covariance function (O'Hagan, 2006).

Here, we use tropospheric ozone columns from SOCOLv3.1 to train the emulator. Interacting contributions to the overall uncertainty in tropospheric column ozone can be identified by comparing the main effect variance (the reduction in the ozone variance when a particular model forcing is fixed, e.g. $NO_x$ emissions), with the total effect variance (the remaining variance in the tropospheric column ozone when everything except a particular model forcing is fixed). Various software packages are available for GP emulation. We used the Gaussian Emulation Machine for Sensitivity Analysis (GEM-SA), available at http://tonyohagan.co.uk/academic/GEM/index.html, to build an emulator for tropospheric column ozone."

15) P6 points 1 and 3: Which type of emissions? Anthropogenic/biomass burning/natural?

We have now noted these in the manuscript – $NO_x$: natural and anthropogenic. CO: natural and anthropogenic. NMVOCs: anthropogenic, biomass burning and biogenic.

16) P6 point 4: I am unclear as to why this is tested. Emissions are included as surface fluxes (i.e. lowest model level) in both SOCOL versions, and to my knowledge, across most models.

This variable was included following the realization that tropospheric ozone in SOCOL is slightly sensitive to the number of levels methane is prescribed on. We were curious as to whether ozone would be similarly sensitive to the number of levels $NO_x$, CO and NMVOCs are prescribed on. The following text has been added to clarify this:

"…variable (5) was included in our analysis to investigate the sensitivity of tropospheric ozone to this implementation. By doing so, we aim to test the exchange of emissions between the boundary layer and free troposphere. The lowermost level in SOCOL covers approximately 100 m, and the 6 lowermost levels combined cover approximately 2.5 km. To explore whether other ozone precursors are sensitive to the number of levels they are prescribed on, variable (4) was included, even though it is prescribed only as a surface emissions flux in most, if not all, CCMs."

17) P7 point 5: I would have thought a priori that the number of levels that methane is prescribed on would not matter for tropospheric ozone amounts, and this is confirmed later in the paper.

Yes – also addressed in point (12) above.

18) P7L24: I am not sure why you would test ranges that are not feasible. E.g. the maximum range for methane (4xCH4) is much larger than even RCP8.5 year 2100 amounts relative to present day. Are we then sure the results of the emulator remain meaningful?

The importance of selecting an appropriate sampling distribution is addressed on P10L17-33 of the ACPD manuscript, and was motivated by observing the "$NO_x$ saturation effect" at scaling factors greater than one. Given the overwhelming dominance of ozone precursors as drivers of tropospheric ozone variability ($\gtrsim$90% in all regions examined), we are confident that, were the analysis to be repeated with a constricted range of scaling factors, the overall results would remain unchanged.

19) P7: The final paragraph explains that physical/meteorological parameters are, by design, not investigated in the emulator experiments. Indeed there could be multiple reasons, besides chemistry, for SOCOL's particularly high ozone bias. This is explained well in the Discussion, but should also be made clear in the Introduction: the methodology used here does not explain (nor is it intended to explain) the entirety of the "remaining ozone bias in SOCOLv3.1" as stated on P3L20.

The following text has been added to the Introduction:

"Our GP emulator experiments have been designed to focus on recent developments surrounding SOCOL's tropospheric chemistry scheme, however the methodology has the potential to be expanded to also include meteorological parameters."

20) P8L2 and Section 3.2: Why not also show results for the global mean tropospheric ozone burden, given its discussion in the Abstract and elsewhere.

We have included the global-mean results in our analysis, and expanded Figures 6 and 8 (now Figures 3 and 5, since the emulator results have been moved to before the CCMI results, following the reviewer's suggestion above) to show the global-mean:

Revised Figure 6 (now Figure 3):

[Figure]

Revised Figure 8 (now Figure 5):

[Figure]

Figure 7 also shows global-mean results, with the corresponding plots for Europe, the US, Asia and Southern Ocean moved to the supplement.

21) P8L12: Reference Morgenstern et al. (2017) who discuss familial relationships between the CCMI models.

Done.

22) P8L22: I do not think you can say ECAM-L90 simulates a "better" representation here since there is no comparison to the observations yet.

This sentence has been relocated to after the following paragraph, once the comparison with observations has been introduced.

23) P9L16: Please provide the ACCMIP MMM global mean tropospheric ozone burden in DU for comparison with CCMI and CMIP5. Also state which, or at least how many, models were considered in the ACCMIP and CMIP mean.

ACCMIP: 30.8 DU calculated from 15 models. CMIP: 30.5 DU calculated from 18 models, as now noted in the text:

"The ACCMIP models simulated, on average, up to 30% more tropospheric column ozone compared with OMI/MLS at northern midlatitudes (Young et al., 2013). The global- annualmean tropospheric ozone column simulated by these models was 30.8 DU, calculated from 15 models. For the 18 CHEM models participating in CMIP5 (those models with interactive chemistry, i.e. ozone was calculated online and not prescribed from a climatology), the climatological-mean annual-mean MMM averaged over 2000-2005 was 30.5 DU (Eyring et al., 2013), which is similar to the MMMs calculated here. The CMIP5 and ACCMIP MMMs also show a stronger interhemispheric gradient than OMI/MLS observations do, consistent with our findings."

24) P9: The CCMI/ACCMIP/CMIP5 comparison is brief. This is fine for the present study, but perhaps the authors could highlight the potential for more detailed future investigation (see also General Comment #2). It would be interesting to see the extent of agreement - or lack thereof - between the different model intercomparisons' simulation of tropospheric ozone, given their different aims and formulations (e.g. a focus on stratosphere-troposphere interactions in the CCMI models vs atmosphere-ocean coupling in CMIP5).

We have included comments on this in the Discussions and conclusions:

"Although ACCMIP, CMIP5 and CCMI all used the same emissions inventories, it is nevertheless interesting that they all produced very similar global-mean 10 tropospheric ozone abundances (approximately 30 DU), given the different foci of the different model intercomparison activities; CCMI focussed on models coupling the stratosphere and troposphere, while CMIP5 focussed on coupling the atmosphere and ocean."

25) Figures 2 and parts of Figure 4, 5: The continuous scale in these figures makes it difficult to distinguish numerical differences between the sub-plots. I recommend a discrete scale as in Figures 3 and 4c, 4f, 5c, 5f.

Changed as suggested.

26) P9L30: Do you mean regionally not globally?

Yes – corrected in the text.

27) P9L33: From Figure 3, it looks like several of the CCMI models also show this bias over the Southern Ocean. Do they share the Wesely deposition scheme?

No, from Morgenstern et al. (2017) they use a variety of schemes – some online, some offline.

28) P10L6: State where this maximum bias occurs.

Done – continental regions in the Northern Hemisphere and Southeast Asia.

29) P10L9, Figure 6: Am I right in thinking that two conditions need to be satisfied in order for the emulator to perform well: having a high R squared value and having the points falling on a 1:1 line? Please clarify.

Yes, and this has now been clarified in the text.

30) P10L10: See earlier comment about using inputs outside feasible ranges, which is acknowledged on P10L30. Do these extremes need to be tested?

We did perform some testing on the extremes, described on P10L17-33 of the ACPD manuscript, and discussed above (point 18).

31) P10L20: Can we explain this? Does it reflect a NOx titration effect?

That is our thinking, yes, and we have added some text to clarify this in the revised manuscript.

32) P10L17, Figure 7: I am a little confused on what to take from this figure: is the "sensitivity" of tropospheric ozone to each parameter determined by the slopes of the sub-plots? If so, why compare the different sensitivities? To determine which parameters are more "important" for tropospheric ozone variability, it makes more sense to compare the variance explained by each parameter (Figure 8). Finally, what does the uncertainty in Figure 7 signify? I may be missing the obvious! Please explain Figure 7 clearly or consider removing.

Figure 7 is useful because it shows whether ozone increases or decreases in response to an individual forcing – this information can't be obtained from Figure 8. Also yes, the slopes can be used to get an indication of how sensitive tropospheric ozone is to a particular forcing. This section has been rewritten (noting that Figure 7 is called Figure 4 in the revised manuscript):

"Figure 4 displays the sensitivity of global-mean tropospheric ozone to each parameter, obtained by averaging over all other parameters, and indicates whether tropospheric ozone increases or decreases in response to an individual forcing/parametrization. Greater uncertainty is indicated where the lines diverge (appearing as a thicker line – i.e., the emulator is less well constrained). Tropospheric ozone exhibits a strong sensitivity to its precursor gases (Fig. 4a-c), and while the correlation between $CH_4$ and CO+NMVOCs is approximately linear, for $NO_x$ there appears to be a saturation effect for scaling factors greater than one, likely due to the "$NO_x$ titration effect" (Thornton et al., 2002)."

33) P10L17: "Figure 7 displays the sensitivity of global-mean tropospheric ozone…" but the figure caption suggests the mean is over the Asian region only.

It should have read that it was for the Asian region in the text. We have now replaced Figure 7 with a plot for the global-mean. Individual plots for Asia, Europe, the US and Southern Ocean are shown in the Supplement.

34) Figure 8: Remove "9 variables" from the figure caption since all 9 variables are not shown.

Done.

35) Figure 8: Could you also show a panel for the global mean burden?

Yes, and now done (shown above, point (20)).

36) Figure 8: Could you explain why the relative importance of CH4 and CO is smaller over Asia than Europe or the US? It would be better to use the same scale on all the panels.

In Europe and the US, the ratio of $NO_x$:CO is high (i.e. there is relatively more $NO_x$ than CO) – see Revell et al. 2015 (www.atmos-chem-phys.net/15/5887/2015/), their Figure 2 and 3d. This would mean that over Asia, where $NO_x$ is relatively less abundant compared with CO (because CO emissions are so large), $NO_x$ would become more important for driving ozone variability. Discussion of this has been added to the text:

"Over Asia, where CO emissions are larger than over Europe and the United States, the ratio of $NO_x$:CO is also lower than it is over Europe and the United States (Revell et al., 2015). $NO_x$ emissions therefore become more important as a driver of ozone variability over Asia (Fig. 5c)."

37) P11L6: "up to 8 DU regionally"

Changed as suggested.

38) P11L12: "up to ~30 DU regionally"

Changed as suggested.

39) Discussions and Conclusions: I very much like this section! I would only conclude with some remarks on the novelty of the emulation technique within this field and its potential future value in the study of ozone biases (see General Comment #2).

This section now concludes:

"Given the results of our multi-model intercomparison as well as previous multi-model studies, our results highlight the need for careful validation of emissions inventories used by global models. However, the way in which emissions are handled by the models also appears to result in biased ozone abundances, and further work is needed to address the challenges of simulating sub-grid processes of importance to tropospheric ozone, in SOCOLv3 as well as in other CCMs. GP emulation may prove a useful tool for such studies, and we have demonstrated its usefulness for understanding tropospheric ozone biases. GP emulation is a powerful tool, and should be considered for use by those wanting to perform detailed sensitivity analyses at low computational cost."

---

## Author Response (AR2)

*Reviewer/editor comments are in **black** and author replies are in blue.*

Thanks to the editor and reviewers for their final comments on this manuscript. Point-by-point responses to each comment are below, followed by a marked up manuscript version.

Laura Revell, University of Canterbury, 14 October 2018.

Co-Editor Decision: Publish subject to minor revisions (review by editor) (11 Oct 2018) by Paul Young

Comments to the Author:

Many thanks to the authors for making the suggested changes by the reviewers. I concur with their assessment that the manuscript should be accepted for publication, subject to responding to the minor revisions that they have outlined. In addition, I have a couple of technical corrections below.

Figs 7 and 8: Please add the units for the RMSE (DU) on the panel titles, as per other figures.

Changed as requested, as well as in the text where appropriate.

Comma splice (e.g., http://bit.ly/2x0j6fW): Replace "..., however..." with "...; however..." (or some other correction) on P4L23, P6L12, P7L2, P11L4, P11L19, P12L30, P13L13 and P14L2. Please check for others - just a bugbear of mine from marking student essays!

These have all been changed; thanks for the grammatical pointers ☺

Report #1 – Anonymous Referee #2

Thanks to the authors for carefully addressing my comments. The manuscript reads well and the emulation procedure is well explained and understandable. I have a few (largely technical) corrections; line numbers refer to the track changed manuscript:

I think this is simply a mistake in the process of track changing – the text under Section 3 (Results) and before Section 3.1, beginning "Figure 7 shows the difference in tropospheric ozone..." should be removed.

Yes, this should have been marked with strikethrough – it has been removed from the final version.

My original point 13) about the tropopause height: My apologies that I was not clear here! I was not asking how the results would change if the tropopause definition was changed from the WMO definition to another. Instead, I was wondering if *inter-model* differences in tropopause height would lead to inter-model differences in the ozone burden. E.g. under the WMO definition, the tropopause height will vary across models due to their different temperature climatologies.

Ah I see. Investigating the influence of tropopause height on the tropospheric ozone burden would be interesting, especially in the context of climate change. With the simulations we look at here, I think it's a little tricky to answer that question. For example, if I compare the global annual-mean tropospheric ozone columns in Fig. 6 with the figure from my earlier response to this comment, I can see that the models don't show a systematic ozone difference. But is it because of the tropopause height, or because of the amount of upper tropospheric ozone simulated by each model? I would guess a combination of both. I agree that this is worth keeping in mind, particularly when it comes to analyzing transient simulations.

P6L1: Keep the removed sentence but change to: "Methane's global average surface mixing ratio is prescribed on the six lowermost model levels."

Changed as suggested.

P7L19: Should be Section 2.4 not 2.5

This has been corrected.

P8L24: I would move the sentence "By doing so, we aim to test the exchange of emissions between the boundary layer and free troposphere." to the end of the paragraph.

Changed as suggested.

P9L26: Change sentence to: "Alongside the global mean, we focus on four regions…"

Changed as suggested.

Report #2 – Referee Edmund Ryan

Thank-you for thoroughly addressing all of my comments. The paper looks great and the description of the emulator + sensitivity analysis parts are much clearer now. Just four really quick things it'd be great if you could change also:

(1) Please change the first sentence of section 2.4 to: "Variance based global sensitivity analysis quantifies the contribution of a single parameter to the variance of a model's output."

Changed as suggested.

 (2) Figure 3. The error bars seem quite large here for a few of the validation points. What have you used for the "emulator uncertainty"? If it's the emulator variance then this is incorrect since the variance is on a different scale (different units) to the mean. If you didn't use the variance then ignore the previous sentence. The only options are the emulator standard deviation (square root of variance, i.e. mean +/- s.d.) or a 95% credible interval (i.e. an error bar whose lower bound is the 2.5th percentile and upper bound is the 97.5th percentile). A 90th credible interval (5th - 95th percentiles) is also fine.

Earlier the emulator variance was shown; this has been replaced with the mean +/- standard deviation, i.e. square root of variance, as advised.

(3) Figure 3 caption: After the word "uncertainty" please include brackets explaining what uncertainty you refer to, e.g. "The error bars indicate the uncertainty (95% C.I.) of the GP emulator output, ..." or the bit in brackets could be "...(mean +/- s.d.) ..." if that's what it refers to.

The caption has been amended to note that the uncertainty is the mean +/- standard deviation.

(4) Figure 4: in your responses to my comments you mentioned that you had changed "scalings" to "inputs" or "parameter" (I can't remember which you said without looking again) since not all of your inputs are scalings (e.g. ELEV). However the x axis labels for figure 4 you still use "scalings". Please can you fix this and also check for other possible general mentions of "scalings" (i.e. okay to refer to an input as a scaling factor if that's what it is but not as a general reference to any of the inputs).

This has been changed to "inputs" as suggested.

[revised manuscript text omitted]